# Bump Feature Detection Based on Spectrum Modeling of Discrete-Sampled, Non-Homogeneous Multi-Sensor Stream Data

Haiyang Lyu, Qiqi Zhong, Donglai Jiao * and Jianchun Hua

Department of Geographic Information Science, School of Internet of Things, Nanjing University of Posts and Telecommunications, Nanjing 210023, China; hlyu@njupt.edu.cn (H.L.); 1022173217@njupt.edu.cn (Q.Z.); b21080612@njupt.edu.cn (J.H.)
* Correspondence: jiaodonglai@njupt.edu.cn

**Abstract:** Roads are the most heavily affected aspect of urban infrastructure given the ever-increasing number of vehicles needed to provide mobility to residents, supply them with goods, and help sustain urban growth. An important indicator of degrading road infrastructure is the so-called bump features of the road surface (BFRS), which have affected transportation safety and driving experience. To collect BFRS, we can collect discrete-sampled, non-homogeneous multi-sensor stream data. We propose a BFRS detection method based on spectrum modeling and multi-dimensional features. With the sampling rate of GPS at 1 Hz and a gyroscope and accelerometer at 100 Hz, multi-sensor stream data are recorded at three different urban areas of Nanjing, China, using the smartphone mounted on a vehicle. The recorded stream data captures a geometric feature modeling movement and the respective driving conditions. Derived features also include acceleration, orientation, and speed information. To capture bump features, we develop a deep-learning-based approach based on so-called spectrum features. BFRS detection experiments using multi-sensor stream data from smartphones are conducted, and 4, 14, and 17 BFRS are correctly detected in three different areas, with the precision as 100%, 70.00%, and 77.27%, respectively. Then, comparisons are conducted between the proposed method and three other methods, and the F-score of the proposed method is computed as 1.0000, 0.6363, and 0.7555 at three different areas, which hold the highest value among all results. Finally, it shows that the proposed method performs well in different geographic areas.

**Keywords:** bump feature of road surface; spectrum modeling of multi-sensor stream data; discrete-sampled and non-homogeneous dataset; movement feature

## 1. Introduction

With the rapid advancement of smart sensors and the influx of crowdsourced data, geospatial information for road surfaces has become increasingly accessible, crucial for the development of smart cities and, more specifically, for maintaining and optimizing road assets [1,2]. Roads, a vital component of urban infrastructure [3–5], endure significant wear due to growing traffic volumes, resulting in various surface irregularities such as bumps, cracks, potholes, and speed breakers—collectively known as bump features of the road surface (BFRS) [6,7]. Effective collection and analysis of BFRS data are essential not only for maintaining road quality but also for ensuring transportation safety and enhancing the driving experience in urban environments [8,9].

Vision-based methods, widely used for their ability to capture road surface conditions through cameras, employ techniques such as computing gradients and Haar features from video frames and images to detect BFRS [10,11]. These methods, alongside advanced machine learning techniques for pothole and crack segmentation [12], and the use of mono and binocular stereo cameras for depth imaging [13–15], form the core of intuitive vision-based BFRS detection. Recent developments in deep learning have significantly improved these detection methods, tailoring them more effectively to the nuanced challenges of BFRS identification [16]. However, these approaches are susceptible to issues like variable lighting, data

noise, and obstructions caused by environmental conditions [10,11]. Alternative approaches using accelerometers and GPS have been developed to monitor road surface anomalies directly from driving data, offering a robust complement to visual methods [9,17]. Furthermore, professional settings increasingly utilize advanced sensor technologies like LiDAR to capture precise geometric details of road surfaces, identifying BFRS through deviations from standard models [18–20]. Additionally, some methods leverage the combination of acceleration sensors, GPS, and auxiliary sensors to capture vehicle vibrations during travel, indirectly detecting BFRS by analyzing unusual sensor data patterns [3,21]. The integration of Internet of Things (IoT) technologies further enhances these methods by facilitating extensive data collection from a network of sensors, thereby creating multi-source datasets essential for addressing complex BFRS detection challenges [3,8,22]. Multi-sensor setups, particularly those incorporating smartphone technologies, offer substantial advantages over traditional methods by enhancing robustness and data diversity. The widespread adoption of smartphones has transformed these devices into powerful tools for crowdsourced data collection on BFRS, which can improve the efficiency of BFRS detection operations and combine user convenience with extensive data coverage [3,7,17,23–25]. Despite these advancements, challenges remain, including (1) managing discrete-sampled, multi-sensor stream data from smartphones in randomly positioned vehicles, (2) limited dataset coverage, and (3) the complexity of detecting and representing BFRS from non-homogeneous datasets collected under varied driving conditions.

To address these challenges and detect BFRS using the widely used motion sensor, this article concentrates on multi-sensor stream data obtained from smartphones positioned in arbitrary poses. The following was carried out: (1) modeling BFRS and augmenting the training dataset with movement features; (2) generating spectral features through fast Fourier transform (FFT) and introducing a deep learning-based BFRS detection method utilizing non-homogeneous, discrete-sampled, multi-sensor data; and (3) representing BFRS using a weighted clustering method. Rather than the numerical measurement of acceleration data from multiple smartphones or varying vehicle types, the contributions of this article are as follows: (1) the modeling of vehicle movement that can deal with the BFRS in different frequencies and driving conditions, along with the augmentation of the training dataset; (2) the representing and detecting of BFRS in account with the noisy sampling; and (3) the presentation and integration of BFRS locations based on the weighted clustering method. To evaluate our approach, we conduct experiments at different geographic areas in the urban area of Nanjing, China. A part of the dataset is used as the training dataset, and others are used as the evaluation dataset; then, comparisons are conducted between the proposed method and three different methods.

## 2. Related Work

As an important factor for transportation safety and driving experience, BFRS detection has long been the heated point for years [3,6,11,16]. Hence, based on the sampling device, related research can be categorized into intuitive vision-based methods and sensor-based methods.

In the intuitive vision-based method, BFRS is collected directly by the camera through images or videos [10,11,13,26]. Images or videos can be collected by specially designed cameras or smartphones during the movement on the road surface. Then, collected images or videos are resampled as a single image with a selected time interval, and then gradients, edges, Haar feature, and some other similar features are computed for BFRS. These image-based techniques are well demonstrated by several recent studies, each employing different methodologies to detect and analyze road surface conditions. Li et al. [27] focused on detecting cracks in road surface images using a Back Propagation Neural Network (BPNN). After successfully identifying the cracks, they utilized image processing techniques to extract the related geometric shapes of each crack, demonstrating the practical application of neural networks in road maintenance. Following a somewhat similar approach but with different tools, Azhar et al. [28] applied histograms of oriented gradients (HOG) to

capture textural features of the road surface. They fed these HOG features into a Naïve Bayes classifier to detect potholes. Once identified, the potholes' geometric information was extracted using the graph-cut image segmentation method, highlighting the integration of machine learning and advanced image processing techniques. Expanding on these methods, Wang et al. [10] computed the Haar feature of each sampled image and trained a classifier using the Adaboost algorithm to detect cracks on the road surface. This approach emphasizes the robustness of feature-based machine learning algorithms in identifying specific types of road damage. Ouma et al. [29] collected RGB images of road surfaces along with the GPS sensor to detect the accurate geometric information of potholes. The RGB image is first enhanced as the gray image, transformed and filtered using the 2D discrete wavelet transformation algorithm, and BFRS are clustered using the fuzzy-c means method. Finally, BFRS are morphologically reconstructed and compared with ground truth objects for validation. In addition, based on photogrammetry or computer vision theory, a mono camera can also detect the generated depth image using the binocular stereo camera. There are various innovative approaches using both image processing and machine learning techniques. Xiao et al. [14] developed a hybrid method that fuses monocular images with LiDAR point clouds to detect road surfaces in diverse scenarios. They utilized classifiers for both images and point clouds based on boosted decision trees and integrated the classification results using Conditional Random Fields (CRFs). Building on the use of imaging for detection, Zhang et al. [13] constructed a pothole detection system based on stereovision. By calculating distances from stereo camera-collected image pairs and utilizing auxiliary GPS data, they were able to precisely determine the size, volume, and position of each pothole, demonstrating the effectiveness of integrating stereo vision with geospatial data. Further exploring stereo imaging, Fan et al. [15] detected potholes using stereo images mounted on a vehicle. They employed random sample consensus (RANSAC) to mitigate the effects of potential outliers during the 3D road surface reconstruction, transformed dense disparity maps to enhance the distinction between damaged and undamaged road areas, and used threshold selection for BFRS detection. These methods underscore the growing influence of deep learning technologies in improving the accuracy of detection systems. Continuing this trend, Kulambayev et al. [16] developed a real-time road surface detection model using a Mask Region-based Convolutional Neural Network (Mask R-CNN), trained and validated with images manually labeled using Microsoft Visual Object Tagging Tool (VoTT). This approach represents a significant leap forward in applying deep learning to road surface assessment, providing high accuracy and real-time processing capabilities. Jenkins et al. [12] proposed pixel-wise road crack segmentation using U-Net, another kind of CNN model, and made a comparison with traditional machine learning algorithms, such as SVM and KNN. A more detailed comparison of vision-based methods can be found at Ma et al. [11], along with some related BFRS-detecting datasets. However, since the collected image or video can be affected by the light, noise samplings, and obstructed sampling, especially in the environment with poor ambient light, or BFRS is blocked on the ground by turbid water and other objects, it becomes difficult to detect BFRS for these vision-based methods.

In the sensor-based method, BFRS is captured using different kinds of sensors, such as LiDAR, accelerometer, gyroscope, GPS, and related auxiliary sensor [9,19–21], with the assumption that abnormal changes caused by BFRS can be recorded during the driving period. One way to collect BFRS is by utilizing the LiDAR sensor, which scans the road surface and detects BFRS based on abnormal changes between the point cloud-fitted surface and typical road surface. Building on the high-accuracy methods described by Siegemund et al. [18], which utilize a 3D point cloud from stereo cameras for detecting and reconstructing curbs and road surfaces, there is a shift in the approach with Sharma et al. [30]. It employed an ultrasonic sensor to monitor road surfaces, using the Dynamic Time Warping (DTW) technique for image processing to match and locate different kinds of BFRS via a GPS sensor. While both methods provide precise measurements of the shape and type of BFRS, they also entail significant costs due to the professional tools required. An alternative approach involves the use of specially designed IoT sensors to

collect BFRS data. Mednis et al. [31] proposed a road surface monitor method using the on-board vehicular sensor system, which was composed of a regular PC computer, low-cost microphone sensor, and GPS receiver. Potholes were detected based on selected thresholds for the microphone sensor recordings. In addition, sensor recordings can be uploaded to the web server through IoT networks [2,4], and BRFS is detected based on multi-source datasets. These methods detect BFRS based on the vibration of vehicles during the driving period, with an acceleration sensor, GPS sensor, and related auxiliary sensor. In recent years, smartphones equipped with multiple sensors have emerged as a popular tool for collecting road condition data. Li et al. [3] utilized this technology to develop a road roughness assessment method. By leveraging the built-in GPS receiver and accelerometer in smartphones, they gathered a spatial series of geo-referenced vertical accelerations of the road surface. They further enhanced this approach by creating a mobile crowdsensing system, which integrates data from multi-source smartphones and a web-based server. This system computes the International Roughness Index (IRI) for each road segment to assess surface quality. Building on this foundation, Rajamohan et al. [8] conducted experiments and developed a prototype system named MAARGHA, designed to estimate the condition and surface type of road surfaces. This work is part of a broader trend in which researchers such as Yi et al. [6], Allouch et al. [32], Mihoub et al. [33], and Mednis et al. [24] have also contributed, each exploring similar methodologies to evaluate and improve road surface assessment using advanced sensor technology and innovative data integration techniques. Zang et al. [34] mounted the smartphone on the bicycle and collected the stream data of the accelerometer and GPS sensor, then BFRS were detected based on abnormal changes using the selected threshold. However, it is the result based on the direct detection from acceleration, which can be affected by the noise of the original recordings. Different from the previous work, Li et al. [7] applied the continuous wavelet transform (CWT) to extract features of acceleration time series using the collected multi-sensor data from smartphones. It took the spatial transformation for the three-axis acceleration dataset, computed the coefficient matrix of CWT, and detected the BFRS based on the threshold of the coefficient. There are also some BFRS detection methods based on multi-sensor data of smartphones that utilize machine learning methods, such as random forest, SVM, KNN [23,25,35], or even the deep learning methods [36,37]. To encode the BFRS feature from collected multi-sensor recordings from smartphones, Chen et al. [17] designed the vibration detection system based on smartphones, accelerometers, and cameras, installed them in the vehicle, conducted the short-time Fourier transform to acceleration, and detected BFRS using the CNN. However, the acceleration recordings were collected at a specific speed with fixed multiple sensors, which limited the feasibility of the application. As described from the above methods, the sensor-based method can collect the actual road conditions and detect BFRS without the limitations of intuitive vision-based BFRS detection methods, with the advantage of multi-sensors and crowd-sourced smartphone datasets. The utilization of non-professional multi-sensor smartphones offers significant advantages, such as accessibility and crowd-sourced data collection, enabling widespread monitoring of road surface conditions. However, the inherent noise in data collection, stemming from varied sensor quality and the randomness of smartphone orientation, presents substantial challenges. These factors lead to non-homogeneous, discrete-sampled stream data that vary under different driving conditions. Feature encoding and detection are crucial in this context as they can effectively handle the uncertainty and variability of data quality, extracting meaningful patterns and insights from the noisy, complex datasets collected by smartphones.

To deal with the non-homogeneous, discrete-sampled multi-sensor stream data from smartphones, the bump feature is analyzed based on the spectrum model; BFRS is detected using different feature channels and is further represented as cluster centers. In addition, the specific deviation and computation of IRI are not going to be conducted in this article, as are the shape and measurement accuracy of BFRS, since the multi-sensor stream data

from the crowd-sourced smartphone device just has limited accuracy and cannot be taken as the replacement of professional surveying tools.

## 3. Methodology

Based on the analysis of related studies, there are intentions for designing the BFRS detection method using smartphones. With the multi-sensor stream data collected at three different urban areas in Nanjing, China, experiments were conducted based on the proposed method. This section introduces the methodological framework and implementation details for BFRS detection. Different from the traditional IoT sensor, it does not extensively describe the data acquisition and measurement process across different sensors or vehicles. Instead, it focuses on acceleration, orientation, and GPS data acquired from smartphones' built-in motion sensors. Here, smartphones, placed in vehicles in arbitrary positions, record the vehicle's location, orientation, and acceleration changes during travel. This allows for the analysis of trajectory records to extract vehicular movement features on the road, such as speed and direction of movement. Subsequently, spectral analysis techniques [38] are employed to represent the spectral features of the acceleration sampling data, which does not rely on direct numerical changes in acceleration to measure BFRS due to inherent sensor noise. Additionally, by simulating different vehicular movement situations based on these movement features, data augmentation is applied to the limited neural network training dataset to enhance the extraction of BFRS and improve the accuracy of the neural network. Finally, the results detected by the CNN [39] are integrated using a mean-shift-based weighted clustering method [40] to represent BFRS.

### 3.1. Motivation and Background

Smartphones currently contain a large array of sensors. Using smartphone sensor data, particularly acceleration data for BFRS detection, is akin to "listening" to road surface changes through an accelerometer, much like how a song records information about its environment. However, unlike voice signals, where a speaker's speed remains relatively stable, vehicle speed and acceleration fluctuate in real-time according to actual road conditions. This variability can cause the same signal to manifest in multiple ways. Consequently, one of the innovations of the proposed method is on analyzing and modeling the relationship between BFRS characteristics and vehicle speed and on enhancing features based on this relationship to accurately capture BFRS information under various speed conditions. The following are relevant for our use case: (1) the GPS sensor to collect two-dimensional location data $P(x, y)$; (2) a gyroscope sensor to collect 3-axis orientations $Ori(o_x, o_y, o_z)$; and (3) the accelerometer sensor to collect 3-axis accelerations $Acc(a_x, a_y, a_z)$. This multi-sensor data are discretely collected at different frequencies and time stamps as a data stream $S$ denoted by (1).

$$S\{P(x, y), Ori(o_x, o_y, o_z), Acc(a_x, a_y, a_z)\} \tag{1}$$

When the vehicle passes an object with the bump of the road surface, abnormal shakes should appear, and they can be recorded by the smartphone placed or mounted in a vehicle. Data stream $S$ captures the state of the road surface. Our goal is to detect BFRS using these data. Challenging issues are that the quality of smartphone sensors varies across devices (denoted as $S_{noise}$), smartphones are arbitrarily placed, and driving behaviors vary (denoted as $D_{speed}$), as denoted by (2). All these conditions lead to a non-homogenous multi-sensor data stream, making it difficult to extract BFRS.

$$BFRS = \text{Detect}(S_{noise}) \; s.t. \; discrete \; \& \; non\text{-}homogenous \tag{2}$$

In this article, smartphones are used as sources for road surface data collection. The sensors within the smartphone are highly integrated and encapsulated within the device. However, compared to professional surveying equipment, the accuracy of sensor data from smartphones is limited. Therefore, to obtain road surface information, the following two issues must be analyzed from both macroscopic and microscopic perspectives:

(1) Smartphone installation/placement in macroscopic aspect: The sensors in smartphones are integrated and fixed on the mainboard, meaning the spatial attitude information of the smartphone is consistent with the accelerometer and orientation sensor data. Thus, it is possible to transform the acceleration records at any moment into an initial posture perpendicular to the road surface along the $z$-axis using the orientation sensor. This means the smartphone can be placed in any orientation within the vehicle as per the method described in this article.

(2) Multiple sensor calibration in microscopic aspect: At the hardware level, the relative positions of the sensors are fixed during integration within the smartphone. At the system level, calibrations can be made through the smartphone's system settings to calibrate between multiple sensors (for example, in iOS, settings can be conducted via "System"->"Privacy"->"Location Services"->"System Services"->"Compass Calibration" and "Motion Calibration and Distance"). Although "data flow/drift" occurs in motion sensors like gyroscopes, where errors accumulate over time, the data preprocessing algorithms provided by smartphone operating systems, such as the "sensor fusion algorithm" in the iPhone's Core Motion framework, can transform this "dirty data" into "clean data". Consequently, the collected dataset, including acceleration and orientation, is preprocessed and filtered. Additionally, before initiating data collection, the smartphone is restarted and left idle for a period to ensure the activation of the device's sensor calibration features. During data collection, the sampling frequency of the smartphone's accelerometer and orientation sensors is consistent, and the impact of the sensors operating at short intervals, such as 100 Hz, on the results is limited compared with vehicle speeds ranging from 0 km/h to 60 km/h. Therefore, linear interpolation techniques have been employed to perform time alignment for the collected data from multiple sensors.

Additionally, the calculation of the smartphone's posture within the sensor array requires reference to electronic compasses, magnetometers, and other sensors. Therefore, during data collection, the smartphone should be kept away from strong magnetic fields to ensure the accuracy of data collection.

Therefore, the first step in our process is temporal alignment for $S$ due to the different time stamps and sampling frequencies. The acceleration timestamp is used as a reference, and stream data from the gyroscope sensor is aligned to the acceleration recordings using linear interpolation. GPS samples are aligned in a similar fashion.

The speed information is computed based on the distance and time duration between consecutive GPS position samplings (cf. Section 3.3 for details). It is assumed that under normal driving conditions (speeds of 0–60 km/h, without sudden traffic accidents, and consistent movement), changes in speed and position over a 1-s interval are continuous. Furthermore, given that GPS signals inherently possess an error greater than 5 m, employing linear interpolation for estimating GPS positions is considered reasonable. Consequently, to correlate acceleration sampling moments with GPS location data, linear interpolation is used to estimate GPS positions at each acceleration sampling moment.

Using the smartphone's recorded 3D orientation information, spatial transformations are performed on the corresponding acceleration data at each moment. This process converts the arbitrarily oriented 3D acceleration data into acceleration information that is parallel to the $xy$ plane and perpendicular to the $z$-axis of the road surface, thereby accurately representing the vertical acceleration relative to the road surface and the horizontal changes in vehicle movement at each moment. The proposed method implements the coordinate system rotation using the Direction Cosine Matrix (DCM) method. At each sampling moment of acceleration, linear interpolation is utilized to temporally align sensor attitude data. Using the three-dimensional posture data corresponding to that moment, a DCM is established. This matrix is then multiplied with the corresponding tri-axial acceleration data to achieve spatial transformation, yielding acceleration data with the $z$-axis perpendicular to the road surface and the $xy$-axis parallel to it. The 3-axis acceleration $Acc$ is affected by vehicle acceleration. Hence, the spatial pose of the accelerometer is reoriented by the timestamp-aligned orientation recordings $Ori_{aligned}$, and the transformed

acceleration $Acc_{trans}(a_x, a_y, a_z)$ consists of a $z$-axis acceleration that is always vertical to the ground and $x$-axis and $y$-axis accelerations parallel to the ground (cf. (3)).

$$Acc_{trans}(a_x, a_y, a_z) = \text{SpatialTransform}\left(Acc, Ori_{aligned}\right)$$
$$with\ a_z \perp ground\ \&\ (a_x, a_y) \parallel ground \tag{3}$$

With the preprocessed multi-sensor stream data $S$, BFRS can be detected through the following steps: (i) preprocess the dataset, (ii) study the relations between driving speed and BFRS, augment the training dataset, (iii) compute the spectrum of BFRS using FFT algorithms, and (iv) extract the specific bump feature according to the modeling result; then, (v) create a deep learning model (CNN), train and validate the model, and (vi) collect BFRS in selected areas; and (vii) finally, represent the BFRS detection result based on the weighted clustering method. The BFRS detection workflow is also depicted in Figure 1.

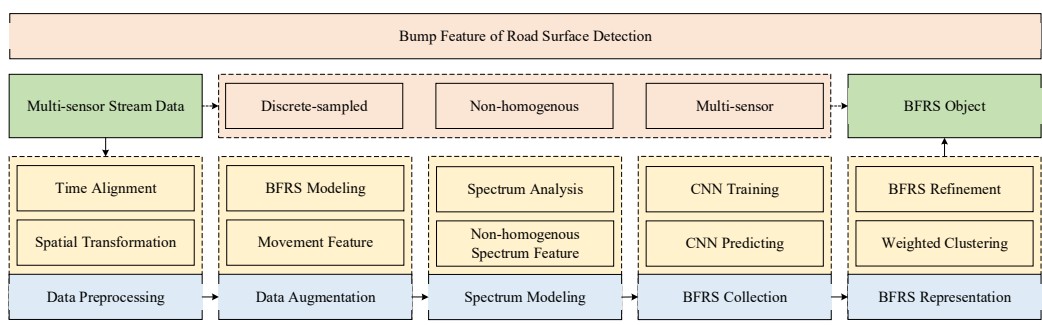

**Figure 1.** The BFRS detection workflow.

### 3.2. Model BFRS and Analyze Driving Behavior

In this work, BFRS is defined as situations/road surface conditions, including speed breakers and transverse gutters, potholes, and cracks distributed on the road surface that cause abnormal changes in the recorded data stream, as shown in Figure 2. Taking a speed breaker shown in Figure 2a as an example, vehicle $A$ is passing the BFRS $B$ with speed $v$. The length (between front wheels and rear wheels) of the vehicle and BFRS is $l_A$ and $l_B$, and the accelerometer sampling frequency is $h$. Hence, the sampling period for object $B$ can be denoted as (4).

$$\Delta t = \frac{l_B}{v} \tag{4}$$

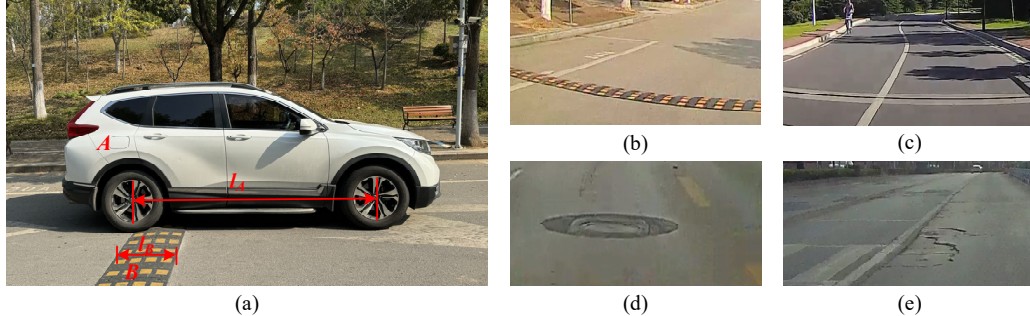

**Figure 2.** The BFRS detection process and different kinds of BFRS. (**a**) The BFRS detection process, (**b**) speed breaker, (**c**) transverse gutter, (**d**) pothole, and (**e**) crack.

However, based on Nyquist's theorem, a continuous signal has a maximum frequency of $f_{max}$, and to fully reconstruct this signal, the sampling frequency $f_s$ must be at least twice the maximum frequency of the signal. Mathematically, it can be represented as $f_s \geq 2f_{max}$. Therefore, the minimum frequency $H$ to catch the bump feature can be denoted as shown in (5).

$$H = \frac{2}{\Delta t} \tag{5}$$

It can be observed that the faster the speed $v$ or the smaller the BFRS length $l_B$, the higher minimum frequency $H$ is needed. For example, in a vehicle with a length $l_A$ = 2.7 m, a length $l_B$ = 0.6 m, and a driving speed $v$ ranging between 1 km/h and 60 km/h, the optimal minimum sampling frequency $H$ should be larger than 0.93 Hz, and the frequency of BFRS ranging between 0.93 Hz and 55.56 Hz, based on (4) and (5).

Accordingly, in this work, the sampling frequency of the accelerometer and gyroscope is set at 100 Hz. On the other hand, suppose a vehicle is driving at the speed $v$ = 30 km/h and the length $l_B$ is 0.6 m, then the BFRS recorded frequency is 13.98 Hz. However, the length $l_A$ between front and back wheels is 2.7 m; there can be a feature with a frequency of 3.09 Hz detected from the recordings since two bump features are recorded by front wheels and back wheels, i.e., the BFRS is recorded twice, and a new signal is generated.

### 3.3. Speed Computation and Data Augmentation Based on Movement Feature

BFRS also causes acceleration vertical to the road surface and a decrease in driving speed. Hence, the acceleration along all 3-axis should be taken into consideration to detect BFRS. The speed is computed using consecutive GPS samples as denoted in (6).

$$v_i = \sqrt{(x_{i+1} - x_{i-1})^2 + (y_{i+1} - y_{i-1})^2} * H_{GPS}/2 \tag{6}$$

In (6), the average speed $v_i$ is taken as the instant speed of $P_i(x_i, y_i)$, and it is computed based on the distance between $P_{i-1}(x_{i-1}, y_{i-1})$ and $P_{i+1}(x_{i+1}, y_{i+1})$, along with the sampling frequency $H_{GPS}$. In addition, since the accelerometer and GPS sampling frequencies are usually different and BFRS are usually detected based on the time stamp of the accelerometer, the computed speed needs to be interpolated to align with the accelerometer time stamp. Here, the linear interpolation method is applied, as denoted in (7).

$$v_k = a(v_k - v_i) + b, \; v_k \in D_{speed} \tag{7}$$

In (7), $v_i$ is the computed average speed at point $Pi$ in (6); $v_k$ is the speed interpolated for time stamp $k$; and $a$ and $b$ are coefficients of the linear interpolation function. When aligning the speed with the accelerometer time stamp, the speed $v_k$ is computed using the precomputed speed $v_i$ and time stamp $k$ in sequence.

The movement feature refers to the speed and direction of the vehicle, which is recorded in the multi-sensor stream data by different driving behaviors. Based on the modeling result and movement feature, recordings containing BFRS are manually selected. However, there are only limited bump features in the training dataset. The intuitive solution is to simulate the driving behavior and augment the dataset based on the movement feature. In our work, two kinds of data augmentation methods, as shown in Figure 3, are used.

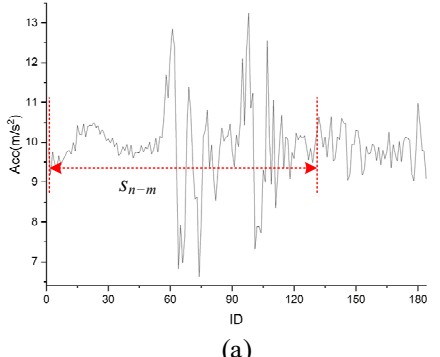

(a)

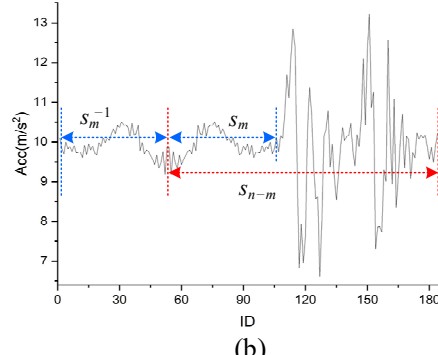

(b)

**Figure 3.** *Cont.*

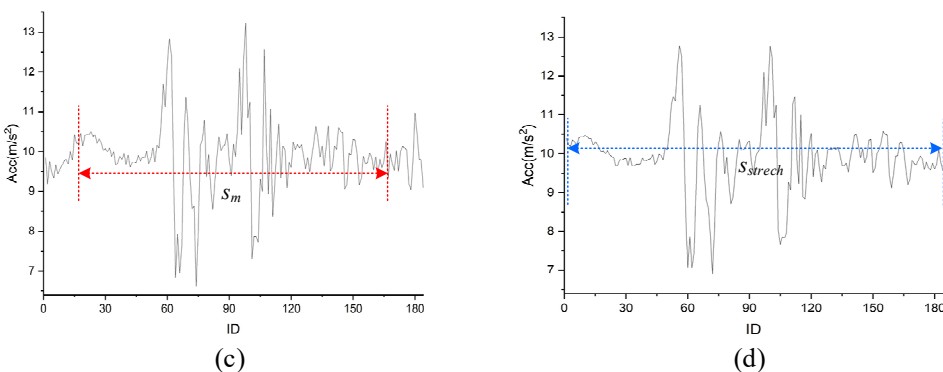

**Figure 3.** Data augmentation based on movement feature. (**a**) To be augmented by driving direction. (**b**) Augmented after driving direction. (**c**) To be augmented by driving speed. (**d**) Augmented after driving speed.

### 3.3.1. Data Augmentation Based on Driving Direction

In the context of BFRS within trajectory recordings, a recording $s$ is composed of $n$ samples, denoted as $s\{r_1, \quad r_2, \quad \cdots \quad r_n \}$, with driving directions indicated by time stamps. BFRS can be sampled at arbitrary positions in these segments due to different labeling methods. Typically, BFRS is positioned at the center of the recording during the manual labeling process. However, various scenarios exist for BFRS labeling, such as labeling at the beginning or end of the recording. To simulate these different scenarios, each recording undergoes a spatial transformation with a random time seed $m \in [0, w]$, and this operation is repeated $2k$ times (with $k$ iterations for pre-resamples and $k$ iterations for post-resamples). To maintain the length of the resampled recordings as the original, the samples on the other side of the resampled recording are repeated and mirrored. For instance, if the recording $s$ is pre-resampled by $n - m$, it is represented as s$\{s_{pre}, s_{n-m}\}$ (as depicted in Figure 3a). However, there is a lack of samples for the previous recordings $s_{pre}$. Thus, the sampling $s_{pre}$ is first represented as $s_m$ and then mirrored as $s_m^{-1}$ to maintain the connectivity of the recordings. The resampled recording $s$ is then denoted as s$\{s_m^{-1}, s_{n-m}\}$ (illustrated in Figure 3b), and the entire process is represented in (8).

$$S = \text{AugmentDirection}(S, m, 2k)\, m \in [0, \ w], k \in N \tag{8}$$

Considering a trajectory where the vehicle moves from $r_5$ to $r_{10}$, the augmentation process involves creating continuity in the sequence. For pre-resample augmentation, if the result lacks initial recordings (e.g., $\{r_2, r_3, r_4\}$), we mirror the existing sequence (e.g., $\{r_5, r_6, r_7\}$) to fill in the gaps. In the case of post-resample augmentation, where the recording extends beyond the original sequence (e.g., $\{r_{11}, r_{12}, r_{13}\}$), the same mirroring technique is applied to maintain continuity. This approach, though simple, ensures the original sequence's continuity and effectively simulates various labeling scenarios. The augmentation process is automated and repeated to ensure a diverse dataset.

### 3.3.2. Data Augmentation Based on Driving Direction

In the BFRS modeling process, it is recognized that vehicles travel at varying speeds, which influences the detection of BFRS based on recordings and movement features. While multi-sensor recordings are captured at specific times and locations, they are inherently logged at a singular speed, omitting the diversity of speed scenarios encountered in real-world driving. To address this limitation, a simulation process that accounts for varying driving speeds is necessary to accurately model BFRS under different speed conditions. Let us assume the vehicle's average speed is $v$, and the number of multi-sensor recordings is $n$. To construct more realistic scenarios, it is essential to simulate speeds $v'$ within a range $v' \in [v_{min}, v_{max}]$. The original recording $s$ is then modified based on the comparison between $v$ and $v'$. If $v > v'$, indicating slower driving conditions, the sampling period of BFRS is extended, and the recording $s$ is stretched. Conversely, if $v < v'$, suggesting faster

conditions, the recording *s* is compressed. These stretching and compressing operations are executed using up-sampling and down-sampling methods, respectively. This augmentation is repeated *k* times with randomly selected speeds $v'$. For instance, when $v > v'$, a scale factor *u* is calculated, and only *m* samples $s_m\{s_{1+m/2}, s_{n-m/2}\}$ are chosen (as depicted in Figure 3c). To maintain the same length as the original recording s, up-sampling is performed through linear interpolation, resulting in $s_{stretch}$ (illustrated in Figure 3d), interpolated with *n* timestamps to match the length of *s*.

$$S = \text{AugmentSpeed}(S, v, v', k) \; v' \in [v_{min}, v_{max}], k \in N \tag{9}$$

In both (8) and (9), the maximum duration *w* for the random seed *m* for spatial transformation is typically set as half the average data length. For example, if the average dataset length is 170, *w* can be set to 80, and the operation repeated 40 times ($k = 20$). Similarly, the randomly selected speed $v'$ ranges from 0 km/h to 60 km/h, with operations also repeated 40 times ($k = 40$). Moreover, while the spectrum feature remains stable during driving direction-based data augmentation, it changes after speed-based augmentation; the frequency of BFRS increases with higher speeds and decreases with lower speeds, as denoted in (4) and (5).

*3.4. Spectrum Modeling for Discrete-Sampled Recordings*

With the analysis result of BFRS, the next step is representing the bump feature based on the Fourier transform and related spectrum. The Fourier transform provides insights into the frequency aspects of signals in the time or spatial domain. It shifts a signal from its inherent domain, typically time or space, to a depiction in the frequency domain. This shift involves breaking down a series of data points into its individual frequency components. The FFT is a computational technique for determining the discrete Fourier transform (DFT) of a dataset or its reverse, as denoted in (10), where $D(k)$ is the output sequence in the frequency domain, $acc(n)$ is the input acceleration sequence in the time domain, *N* is the number of sample points, and *k* is the frequency index, ranging from 0 to $N - 1$.

$$D(k) = \sum_{n=0}^{N-1} Acc(n)e^{-j(2\pi/n)kn} \tag{10}$$

Analyzing the spectral components focuses on understanding the frequencies present in sampled datasets, and this mathematical equation switches a signal from its original time or spatial representation to a version defined by frequency. According to (10), the spectrum *Sf* is computed using sliding window w with overlap rate *e* that moves along the time dimension and computes FFT in a short time period. In this article, the spectrum is applied to represent the bump feature from spatially transformed $Acc_{trans}$, as denoted in (11).

$$Sf = \text{SpectrumAnalysis}(Acc_{trans}, l_w, o_w, H_{bfrs}) \tag{11}$$

Since the abnormal change in BFRS is recorded twice by the front wheel and back wheel of vehicles, as depicted in Figure 2, the sliding window length $l_w$ of the BFRS to generate spectrum is set as 170, which can cover the average abnormal change under the speed condition $v \in [1, 60]$ km/h according to (4). Moreover, the frame duration is set as 50, the hop duration is set as 5, and the specific frequency $H_{bfrs}$ of BFRS ranges from 3 to 50 Hz, based on the BFRS modeling result in Section 3.2. Assumption that the original acceleration recording is depicted in Figure 4a–c, the spectrum computed based on FFT is depicted in Figure 4d–f. It can be observed that abnormal changes in the original recordings are distributed along the time direction from the spectrum view, and the feature of the abnormal change can be represented by frequency domain. The heavier changes in the acceleration, the brighter (represented by the yellow color, in respect to the blue color) the color in the frequency domain. Moreover, compared with one-dimensional changes in the original acceleration recording in the upper row, abnormal changes in the low row are depicted just like the magnifier in the frequency domain, and the directional non-homogenous feature can be more clearly represented.

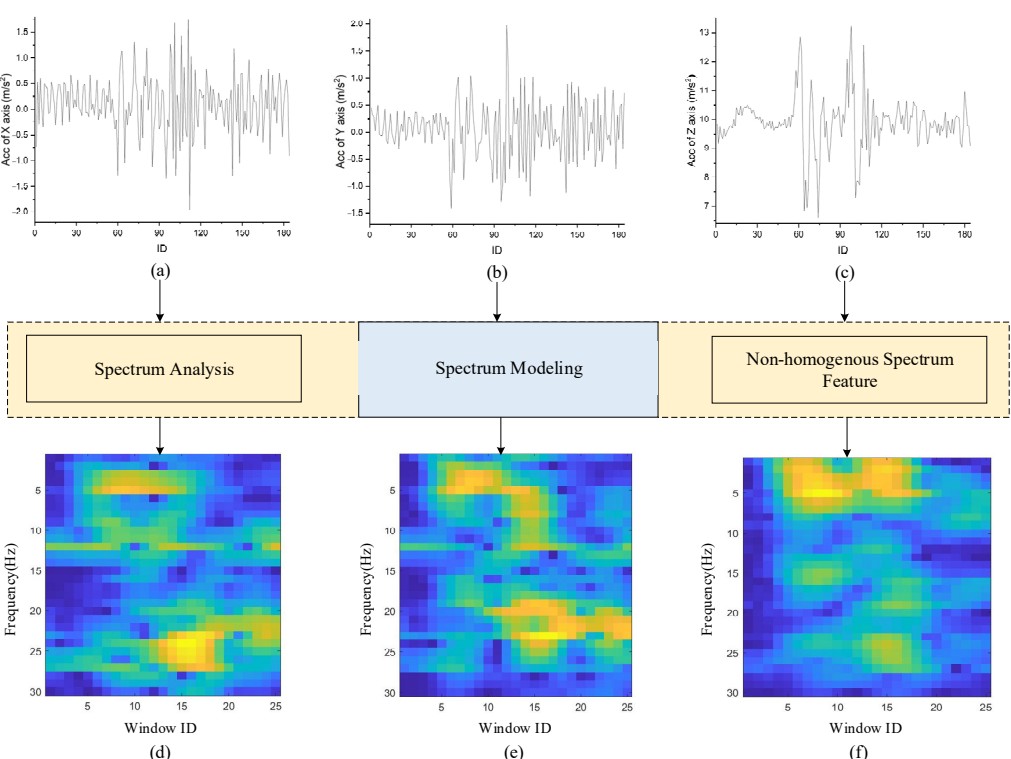

**Figure 4.** The non-homogenous spectrum features in different dimensions. (**a**) Acceleration on *x*-axis. (**b**) Acceleration on *y*-axis. (**c**) Acceleration on *z*-axis. (**d**) Spectrum feature on *x*-axis. (**e**) Spectrum feature on *y*-axis. (**f**) Spectrum feature on *z*-axis.

### 3.5. BFRS Collection Using Non-Homogeneous Spectrum Feature

As analyzed in the previous section, the abnormal changes can be observed via the *z*-axis acceleration records, while there are also similar abnormal change features in the *x*-axis and *y*-axis acceleration recordings. Therefore, the spectrum for 3-axis accelerations is computed based on (7), and four-dimensional spectra are generated, as denoted in (12).

$$\left(sf_x, sf_y, sf_z\right) = \text{MultiSpectrum}\left(Acc_{trans}\left(a_x, a_y, a_z\right)\right) \tag{12}$$

With these multi-dimensional spectra, BFRS are initially detected based on the CNN model. As a type of feed-forward neural network that learns feature engineering by itself via filter (or kernel) optimization, CNN can automatically learn meaningful features without manual engineering. By training on vast amounts of data, CNN can learn a hierarchy of features, from simple to complex, and achieve or surpass human-level performance in various tasks. Due to its exceptional performance in computer vision tasks such as image classification and object detection, CNN has become a standard technique in this domain, and it is particularly well-suited for processing matrix data. Hence, the CNN model is applied in this article to deal with the multi-dimensional spectrum data. The CNN model usually consists of the convolutional layer, activation function, pooling layer, and fully connected layer, as follows:

(1)  A convolutional layer uses a set of learnable filters (or kernels) to scan input.
(2)  The batch normalization layer normalizes the data on each mini-batch to accelerate the model.
(3)  The activation layer is used to introduce non-linearity, with the ReLU function usually employed.
(4)  The pooling layer down-samples the input data to reduce the data's dimensionality.
(5)  The dropout layer turns off a subset of neurons at random during training to prevent overfitting.

(6) A fully connected layer is used for tasks like classification at the end of a CNN.

(7) The softmax layer converts a set of values into a set between 0 and 1, the sum of which equals 1.

(8) A classification layer is used for classification tasks.

In this article, a simple CNN model is designed and applied with 5 convolutional layers, 5 batch normalization layers, 5 activation layers, 4 pooling layers, 1 dropout layer, 1 fully connected layer, 1 softmax layer, and 1 classification layer. The 3-axis acceleration data are initially encoded as a spectrum feature in dimensions of $30 \times 25 \times 3$ and converted to the size of $30 \times 25 \times 12$ by the first round of convolution operations, followed by the batch normalization. Then, the process is repeated for the other four times, with the dimensions becoming $15 \times 13 \times 24$, $8 \times 7 \times 48$, $4 \times 4 \times 48$, and $4 \times 4 \times 48$, followed by the dropout, fully connection, and softmax operations with dimension finally becoming $1 \times 1 \times 2$. Through these operations, the CNN model can "hear" the voice of the road surface and detect BFRS from the input dataset. The detailed structure is depicted in Figure 5.

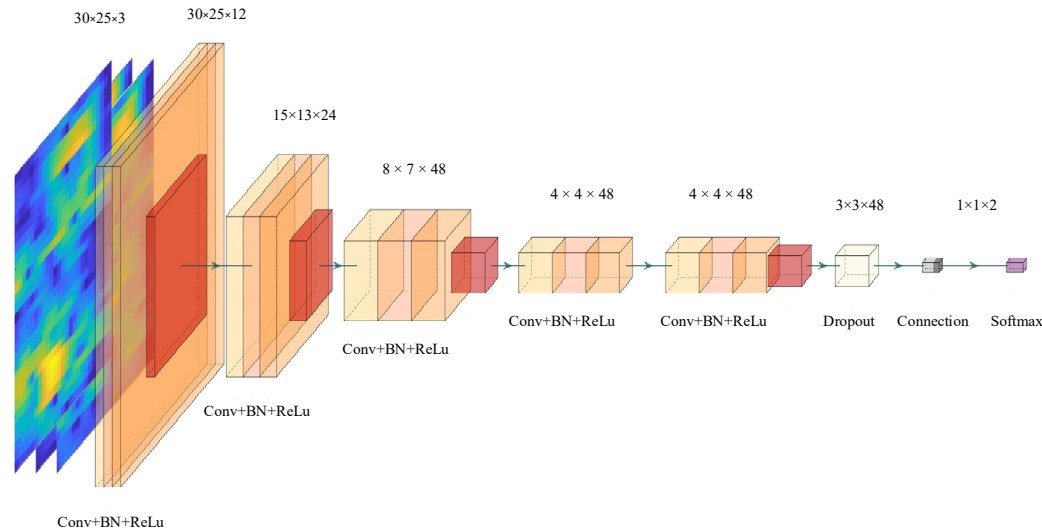

**Figure 5.** The BFRS detection architecture.

In Figure 5, the input three-dimensional spectra are input into the CNN model, followed by a convolutional layer with bath normalizations and an activation layer (using the ReLU function), a pooling layer (using the max pooling function), then the operation is repeated for another three rounds; finally, the BFRS are generated by the full connected layer with dropout and softmax functions. In addition, some complex CNN models can also be applied since they are just a module of the BFRS detection framework. After the training and predicting process, the $BRFS_{info}$ can be collected, as denoted in (13).

$$BFRS_{info} = \mathrm{CNN}\left(sf_x, sf_y, sf_z\right) \tag{13}$$

*3.6. BFRS Representation Based on the Weighted Clustering*

After the processing of the CNN model, the labels of BFRS and probability are collected as $BFRS_{info}\{BFRS_{label}, BFRS_{prob}\}$, while there is still noise and isolated detection results that have inadequate confidence and cannot be taken as proper detections. In addition, the bump feature for the same BFRS can be different, and how to deal with the non-homogeneity remains a challenge in the traditional signal processing operation. To solve the problem, the detection result is first refined based on the buffered windows that keep the consecutive confidence of the detection result. After the process, each BFRS is labeled as a segment of the original recording; hence, the second step is representing the BFRS as a single position according to the weighted clustering. The refinement and clustering process can be depicted in Figure 6.

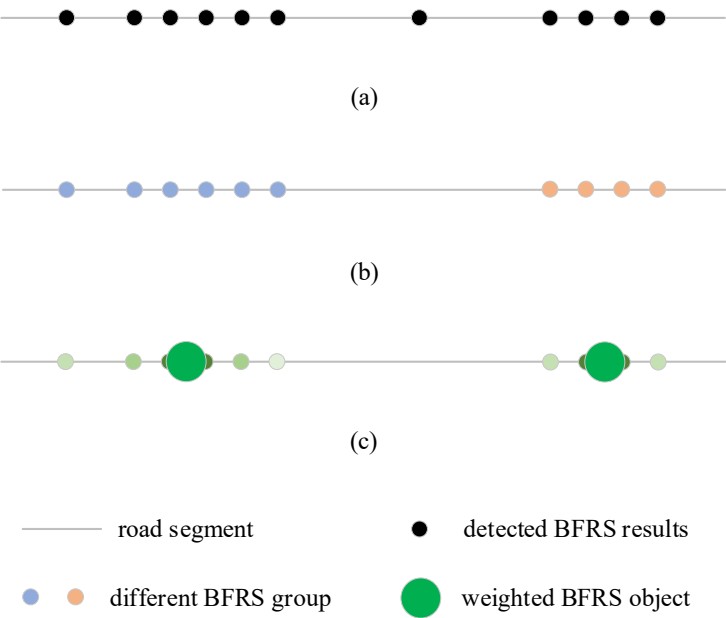

**Figure 6.** The BFRS representation from segment to position. (**a**) BFRS detected by CNN. (**b**) Refined and segmented as each cluster group. (**c**) Represented by weighted cluster center.

### 3.6.1. Refinement Using the Buffered Window

Suppose the detected result is subtracted with a buffered window $w_{buf}$ that slides from the start time to the end of the recording, which can be denoted as $w_{buf}\{BFRS_{info}^{i}, BFRS_{info}^{i+1}, \cdots, BFRS_{info}^{i+n}\}$. Then, the refinement process is conducted for each result in $w_{buf}$. The detected result in $w_{buf}$ is judged as BFRS under the following three conditions:

(1) More than half of the detected result $BFRS_{label}$ is not labeled as "road".
(2) More than 60% percent of detected results in the buffered window $w_{buf}$ is BFRS.
(3) The probability $BFRS_{prob}$ of a detected result is larger than the probability threshold $Prob_{min}$.

The point in the buffered window is relabeled as "true" and taken as the right detected BFRS, or the detected is taken as "false", and the operation can be depicted in (14).

$$BFRS_{refined} = \text{Refine}\left(BFRS_{info}, w_{buf}, Prob_{min}\right) \tag{14}$$

In (14), the length n of the buffered window $w_{buf}$ is usually set as 10, or some other larger value that with the high confidence of the detection result, and the probability threshold $Prob_{min}$ is set as 0.7, which is a little higher than the probability detected by the CNN model, as depicted in Figure 6a.

### 3.6.2. Representation Using Weighted Clustering

With the refinement process, the detected result $BFRS_{refined}$ is converted to each segment $BFRS_{group}$, which is composed of consecutive points $BFRS_{group}\{b_1, b_2, \cdots, b_n\}$. However, the BFRS is the specific position, such as the speed breaker or pothole, so the next step is to distinguish the segments $BFRS_{group}$ from each other and convert the consecutive segment into a specific object location $BFRS_{obj}$, as depicted in Figure 6b. The operations are as follows:

(1) Compute the difference index of each detected sampling point based on the index orders from the original recordings.

(2) Segment detected sampling points into each group $BFRS_{group}$ if the index difference is within the threshold value $d$, as denoted in (15).

$$BFRS_{group} = \text{Cluster}\left(BFRS_{refined}, d\right) \quad (15)$$

(3) Each detected sampling point in $BFRS_{group}$ is composed of the specific coordinate $P_i(x, y)$ and the related probability $BFRS_{prob}^i$; hence, points in each $BFRS_{group}$ can be further integrated into a single point $BFRS_{obj}$ (depicted in Figure 6c) based on the weight of $BFRS_{prob}^i$, as denoted in (16).

$$BFRS_{obj} = \text{Weight}\left(BFRS_{group}, BFRS_{prob}\right) \quad (16)$$

In (15), there is a key parameter d that segments the consecutive detected sampling point into each group, and it can be selected depending on the data quality of the original recordings. Usually, d can be set as 3 if too many noisy samplings are not included. While in (16), the coordinate of $BFRS_{obj}$ in each $BFRS_{group}$ is automatically computed based on each detected sampling point and the related $BFRS_{prob}$ (depicted as the graded green color in Figure 6c), and no other parameters are needed.

### 3.7. BFRS Detection Algorithm from Multi-Sensor Stream Data

The BFRS detection framework consists of the following four steps: Firstly, preprocess the collected discreet multi-sensor stream data from smartphones, including (1) timestamp alignment for multi-sensor recordings based on linear interpolation and (2) spatial transformation for the 3-axis acceleration data. Secondly, augment the training dataset based on the movement feature, model BFRS using the driving speed and 3-axis acceleration recordings, and represent the bump features based on spectrum modeling. Thirdly, design the CNN-based deep learning network to deal with the non-homogenous BFRS spectrum and collect BFRS. Finally, represent the BFRS detection result based on the weighted cluster method. The algorithm is depicted in Algorithm 1.

---

**Algorithm 1** The BFRS detection algorithm

---

Input : $S_{noise}$ denoted as (1) and (2)
Output: $BFRS$ denoted as (2)
BFRSDection($S_{noise}$) $\left\{ P(x,y), Ori(o_x, o_y, o_z), Acc(a_x, a_y, a_z), D_{speed} \right\}$
//step 1: preprocessing and spatial transformation
$Ori_{aligned} = \text{AlignOrientation}\left(Ori(o_x, o_y, o_z), t_{acc}\right)$; //Align the Orientation with Acceleration
$Acc_{trans}(a_x, a_y, a_z) = \text{SpatialTransform}(Acc(a_x, a_y, a_z), Ori_{aligned})$; //based on (3)
//step 2: data augmentation and BFRS spectrum modeling
$H_{bfrs} = \text{BFRSModeling}(S_{noise}, D_{speed})$ //based on (4) and (5)
$S\{P, Acc\} = \text{AugmentDataset}(Acc_{trans}, D_{speed})$ //based on (6)–(9)
$Sf(sf_x, sf_y, sf_z) = \text{ComptueSpectrum}(S, H_{bfrs})$; //based on (10)–(12)
//step 3: BFRS collection based on CNN
$para_{CNN} = \text{TrainCNN}(Sf)$;
FOREACH $Sf_i$ IN $Sf$
    $W_i = \text{SildingWindow}(Sf_i)$; //slice the spectrum
    $BFRS_{info}^i\{BFRS_{label}, BFRS_{prob}\} = \text{PredictCNN}(W_i, para_{CNN})$; //based on (13)
END
$BFRS_{refined} = \textbf{BFRSRefine}(BFRS_{info}, w_{buf}, Prob_{min})$ //based on (14)
//step 4: BFRS representation based on weighted clustering
$BFRS_{group}\{P_{BFRS}, BFRS_{prob}\} = \text{Clustering}(BFRS_{refined}, d)$; // based on (15)
$BFRS_{obj} = \text{WeightedIntegration}(BFRS_{group})$; // based on (16)

---

## 4. Experiments and Discussion

This section focuses on the performance of the BFRS detection method. The multi-sensor stream data are acquired using smartphones with GPS, gyroscope, and accelerometer sensors. GPS locations, three-axis orientations, and three-axis acceleration datasets are collected. Smartphones are used as the source for sensor data collection, utilizing the inherent data processing techniques of the smartphone operating system. This setup effectively addresses the issue of "data flow/drift" in motion sensors (such as the gyroscope). The raw acceleration data are preprocessed using these techniques before being recorded by the MATLAB Mobile 2023 app and subsequently sent to the proposed method. Each dataset collected typically lasts no longer than 10 min. Due to the robust preprocessing capabilities of the smartphone, extra calibration for the motion sensors is not conducted during the data collection process. BFRS is modeled after timestamp alignment between the accelerometer and GPS and gyroscope sensors. Then, the direction of the three-axis accelerometer is reoriented by the three-axis acceleration information, and the spectrum model is applied to the accelerometer data. The BFRS is computed based on CNN and represented using the weighted cluster method. After that operation, the cluster result is compared with the actual BFRS positions, treating detections within 10 m of the actual BFRS as correct and those outside this range as wrong. These criteria serve as our evaluation metrics, and precision, recall, and F1-score are calculated. Based on the criteria, comparisons are conducted with different BFRS detection methods; in addition, BFRS in different areas is computed, and results are evaluated with different methods.

### 4.1. Multi-Sensor Stream Data Collected Using Smartphone

As shown in Figure 7, the total length computed based on GPS recordings is 6411 m with speed ranges from 0 km/h to 39.65 km/h, in an area of 0.92 km$^2$. The time duration of the stream data is 860 s, with 860 sampling of GPS points and 86,662 records of three-axis acceleration and three-axis orientation records. In the research area, BFRS refers to surface features that have variations in elevation greater than 3 cm and extend horizontally along the road for more than 4 cm. Due to the geographic coordinates inherent in the detection results, which differ from traditional methods of evaluating neural network models, data from a specific spatial area was selected as the training set, and data from a different area (not included in the training set) was used for testing. In Area A, the dataset is divided into a training dataset (outside the black dotted rectangle area) and a test dataset (inside the black dotted rectangle area, i.e., Area A), and the training dataset is used to train and validate the CNN model, while the test dataset is used to make validation and comparison with other related methods. The detailed statistics of the collected multi-sensor stream data are depicted in Table 1.

**Table 1.** Statistic of collected multi-senor stream data.

|  | Length | Duration | GPS Records | Acceleration Records | Orientation Records |
|---|---|---|---|---|---|
| Area A | 6411 m | 860 s | 860 | 86,662 | 86,662 |

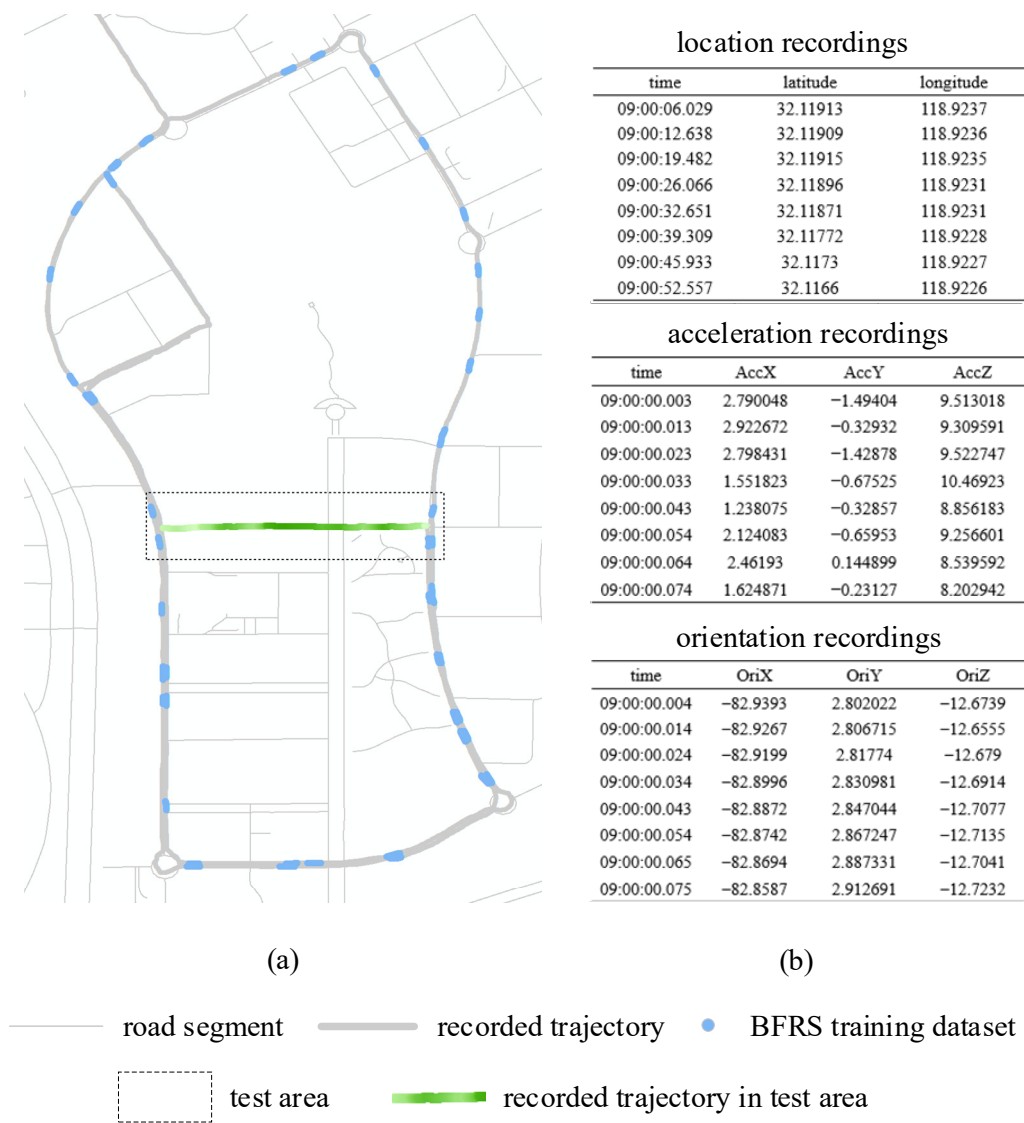

location recordings

| time | latitude | longitude |
| --- | --- | --- |
| 09:00:06.029 | 32.11913 | 118.9237 |
| 09:00:12.638 | 32.11909 | 118.9236 |
| 09:00:19.482 | 32.11915 | 118.9235 |
| 09:00:26.066 | 32.11896 | 118.9231 |
| 09:00:32.651 | 32.11871 | 118.9231 |
| 09:00:39.309 | 32.11772 | 118.9228 |
| 09:00:45.933 | 32.1173 | 118.9227 |
| 09:00:52.557 | 32.1166 | 118.9226 |

acceleration recordings

| time | AccX | AccY | AccZ |
| --- | --- | --- | --- |
| 09:00:00.003 | 2.790048 | −1.49404 | 9.513018 |
| 09:00:00.013 | 2.922672 | −0.32932 | 9.309591 |
| 09:00:00.023 | 2.798431 | −1.42878 | 9.522747 |
| 09:00:00.033 | 1.551823 | −0.67525 | 10.46923 |
| 09:00:00.043 | 1.238075 | −0.32857 | 8.856183 |
| 09:00:00.054 | 2.124083 | −0.65953 | 9.256601 |
| 09:00:00.064 | 2.46193 | 0.144899 | 8.539592 |
| 09:00:00.074 | 1.624871 | −0.23127 | 8.202942 |

orientation recordings

| time | OriX | OriY | OriZ |
| --- | --- | --- | --- |
| 09:00:00.004 | −82.9393 | 2.802022 | −12.6739 |
| 09:00:00.014 | −82.9267 | 2.806715 | −12.6555 |
| 09:00:00.024 | −82.9199 | 2.81774 | −12.679 |
| 09:00:00.034 | −82.8996 | 2.830981 | −12.6914 |
| 09:00:00.043 | −82.8872 | 2.847044 | −12.7077 |
| 09:00:00.054 | −82.8742 | 2.867247 | −12.7135 |
| 09:00:00.065 | −82.8694 | 2.887331 | −12.7041 |
| 09:00:00.075 | −82.8587 | 2.912691 | −12.7232 |

(a)                                          (b)

—— road segment  ——— recorded trajectory  • BFRS training dataset

⬚ test area  ▬ recorded trajectory in test area

**Figure 7.** The research area and related multi-sensor stream data. (**a**) Research area. (**b**) Related multi-sensor recordings.

### 4.2. Non-Homogeneous Analysis and BFRS Modeling

The smartphone is arbitrarily mounted or placed in the vehicle, but there is an extensive abnormal change in the direction vertical to the road surface, and the spatial transformation is necessary to convert the $z$-axis acceleration into the vertical direction of the road surface. With timestamp-aligned three-axis acceleration and three-axis orientation recordings, the real-time spatial transformation is applied to the three-axis acceleration. The spatial transformation for the acceleration stream data is depicted in Figure 8.

As shown in Figure 8, there is an obvious change in the three-axis acceleration before and after the spatial transformation. In Figure 8a, both $x$-axis, $y$-axis, and $z$-axis accelerations change during the sampling period, and the average acceleration for the three axes is 2.2278 m/s², −0.6017 m/s², and 9.4296 m/s², respectively. In Figure 8b, the average $x$-axis and $y$-axis acceleration becomes 0.0513 m/s² and −0.0533 m/s², which is consistent with the actual case (as the vehicle moves forward at a nearly constant speed), and the $z$-axis acceleration is to 9.7075 m/s²; however, abnormal changes are concentrated on the $z$-axis, while there are also related slight changes in the $x$-axis and $y$-axis. It shows strong relations with BFRS after spatial transformation. Additionally, it can be observed that BFRS impacted both the vertical ($z$-axis) and horizontal ($xy$-axis) directions of the vehicle. In

the *z*-axis direction, the changes in acceleration are quite noticeable, typically fluctuating above $\pm 3$ m/s$^2$. In the *x*-axis direction (assuming it is the direction of vehicle travel), a noticeable deceleration effect is also presented, although the changes in acceleration were relatively minor. However, employing a threshold method to determine BFRS can lead to significant errors due to the inherent noise in acceleration recordings. Therefore, the proposed method utilizes spectral features to detect BFRS, which also aids in mitigating the impact of acceleration noise to some extent.

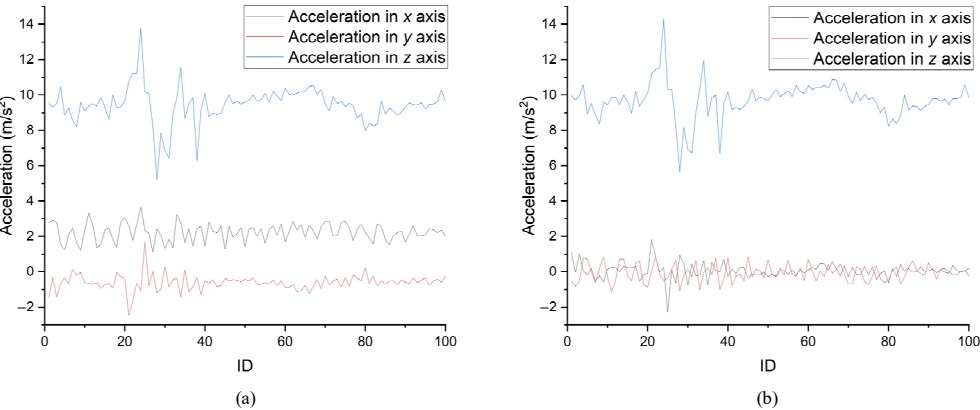

**Figure 8.** Spatial transformation for the acceleration data. (**a**) Before spatial transformation. (**b**) After spatial transformation.

BFRS is modeled with the relations between the geometric shape and driving speed in this research, and the BFRS training dataset is augmented based on the modeling result since the size and quality of the training dataset have a direct effect on the neural network. To have an intuitive visualization of the augmented results, the experiment is conducted on the actual collected recordings using the proposed augmentation method, as depicted in Figure 9. In the experiment, the spectrum of BFRS is computed using the FFT algorithm based on (10)–(12), with the parameter set as Section 3.3.

As seen from the experiments in Figure 9, the spectra of BFRS are computed and visualized for the acceleration of the *x*-axis, *y*-axis, and *z*-axis. In Figure 9a–c, it is the comparison result of data augmentation by driving direction, and the original recording in Figure 9b is pre-resampled with 33 samplings, while post-sampled with 41 samplings. In Figure 9d–f, it is the comparison result of data augmentation by driving speed; the original recording in Figure 9e is speed increased with 123% of the original speed in Figure 9d, while decreased with 85% of the original speed in Figure 9e. In Figure 9a–c, the spectrum feature keeps steady during the pre-resample and post-resample processes. However, in Figure 9d,e the same spectrum features vary under different speed conditions; the spectrum feature representation of the same BFRS is stretched or squeezed by different vehicle speeds and becomes non-homogenous. In addition, to validate the effectiveness of the data augmentation method, two different spectrum feature maps are generated based on the actual acceleration recording under speed conditions of 18 km/h and 27 km/h, as depicted in Figure 10.

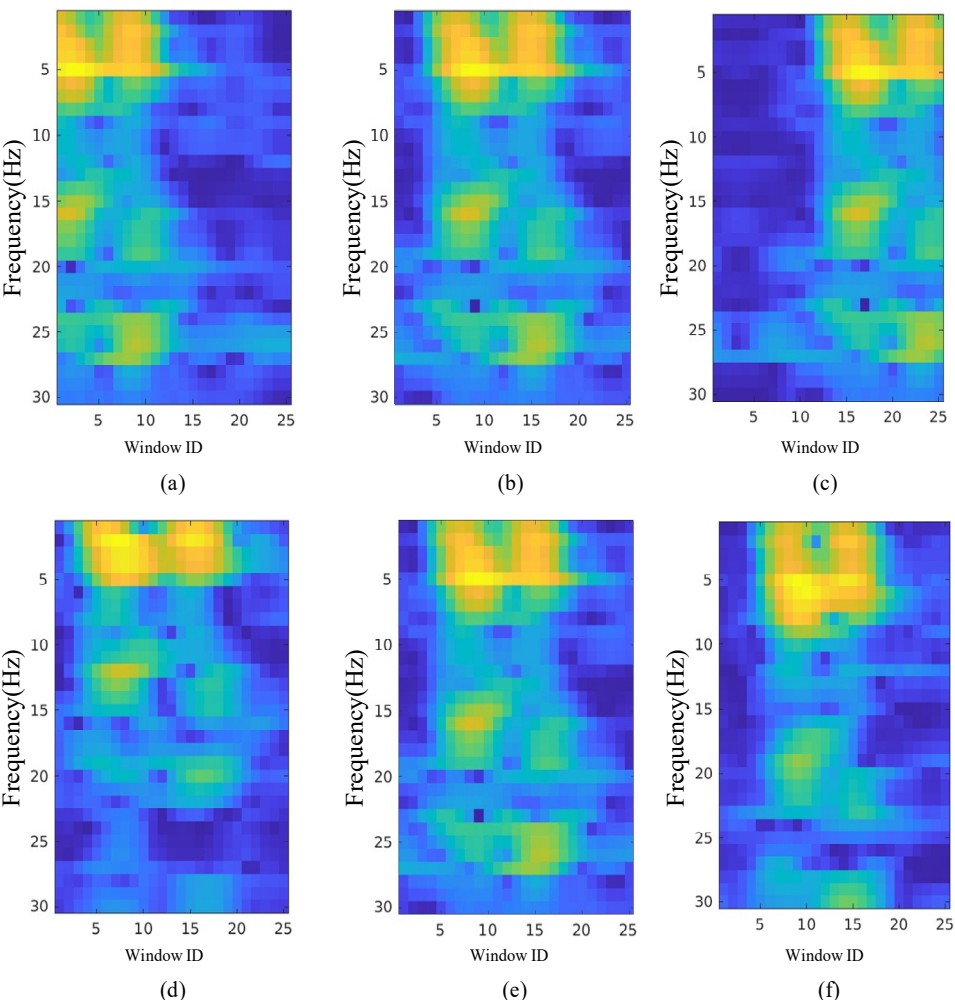

**Figure 9.** Comparisons of spectrum modeling results. (**a**) Augmented by driving direction: pre-resample. (**b**) Augmented by driving direction: original dataset. (**c**) Augmented by driving direction: post-resample. (**d**) Augmented by driving speed: low speed. (**e**) Augmented by driving speed: original dataset. (**f**) Augmented by driving speed: high speed.

In Figure 10, the actual acceleration recording of the same kind of BFRS is collected under different speed conditions. There is a strong saliency (the light-yellow color in Figure 10b) in the lower frequency domain when recording under the speed of 18 km/h in Figure 10a; however, the strong saliency (the light-yellow color in Figure 10d) moves to the higher frequency domain when recording under the speed of 27 km/h in Figure 10c, which is consistent with the assumption analyzed in Section 3.3. In addition, the augmentation result based on the speed direction is not further experimented with, as the spectrum feature keeps steady during the process. Despite the presence of noise in the original recordings, the method transforms one-dimensional acceleration changes into two-dimensional spectral features. By distinguishing BFRS characteristics from noise across different frequency spectrum coefficients, the proposed method effectively reduces the impact of noise on experimental results. Based on the proposed data augmentation, there are non-homogenous feature representations under different speed conditions, and the feature can be extracted and strengthened based on the spectrum modeling method, which would contribute to the classification of BFRS in the next section.

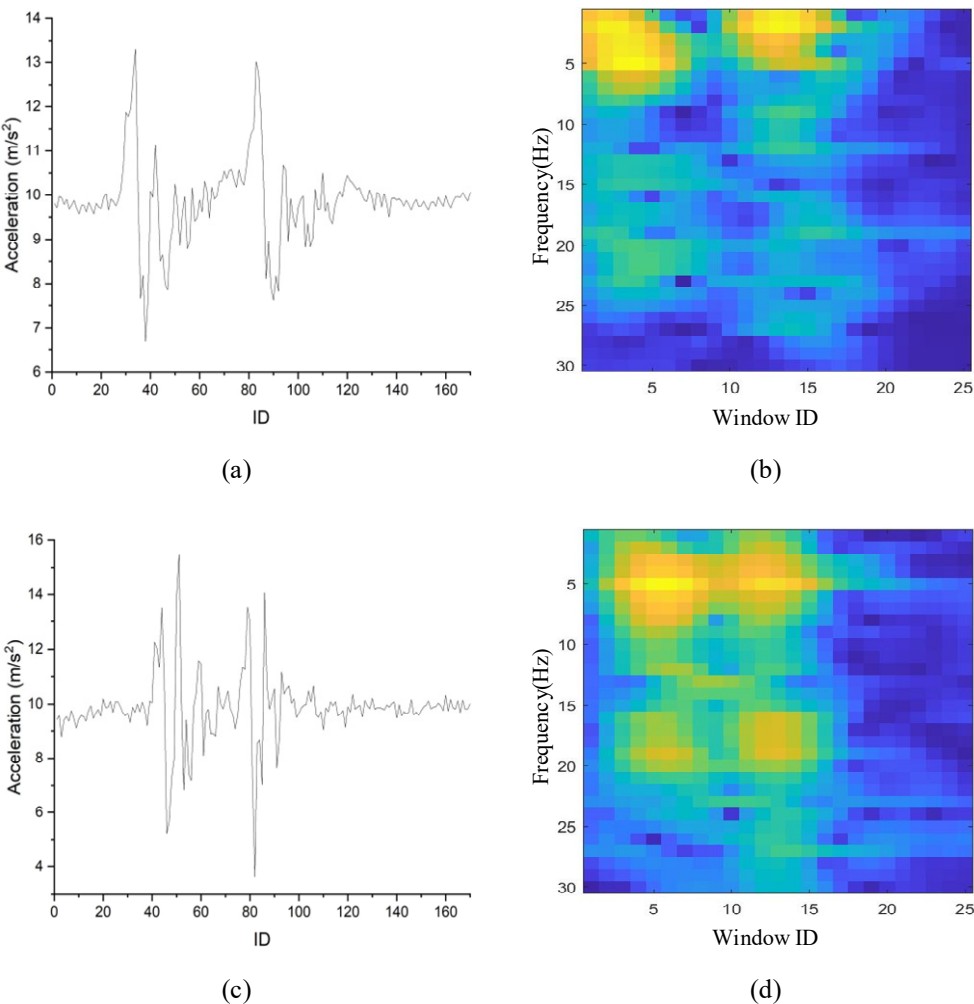

**Figure 10.** The actual acceleration recording and related spectrum feature. (**a**) Acceleration recording with actual speed 18 km/h. (**b**) Related spectrum feature of 18 km/h. (**c**) Acceleration recording with speed 27 km/h. (**d**) Related spectrum feature of 27 km/h.

### 4.3. BFRS Detection and Representation

Since BFRS is affected by the abnormal changes in the three-axis acceleration, all the spectra in three dimensions are computed and taken into the CNN model. In addition, the non-homogenous is caused by different driving speeds, as discussed in the previous section. The CNN model to detect BFRS is designed according to Section 3.3, with the spectrum in the three-axis acceleration. The training parameters are set as follows: {number of filters: 12; mini batch size: 50; learning rate: 0.0001; max epochs: 30; dropout probability: 0.2; padding mode: same}, with other parameters the same as Section 3.5. During the training process, the dataset is divided into the following two categories: 80% of datasets are taken as training datasets, and 20% datasets are taken into validate datasets. Furthermore, to distinguish the BFRS from the normal road features, 166 segments of normal road segments are randomly selected in the research area and further labeled as "background". During the training process, the detection result is validated after each epoch, and the CNN model converges. Then, the BFRS detection operation is conducted on the test dataset of Area A, with the trained learnable parameters. The multi-dimensional spectrum features of acceleration recordings are sent to the CNN model, output with the detected labels, segmented and refined as consecutive segments, and weight-clustered and represented as BFRS, as depicted in Figure 11.

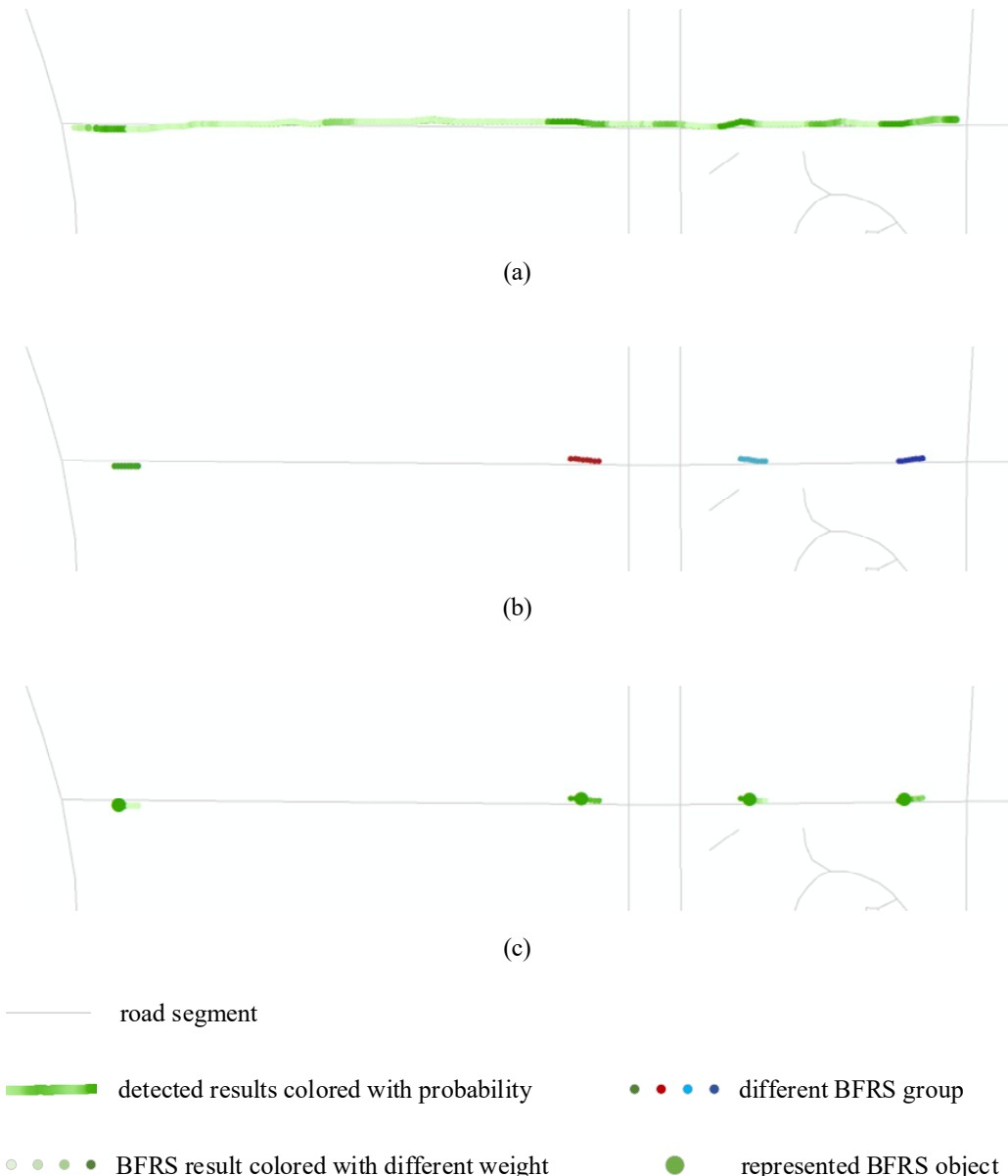

Figure 11. BFRS detection results. (**a**) BRFS detected results colored with probability. (**b**) Segmented and refined as different BFRS groups. (**c**) Represented by weighted cluster center.

In Figure 11a, the BFRS is initially detected based on the CNN model; each recording is assigned the probability of BFRS and further visualized by the graded color in greed (the darker the point, the higher the probability of the BFRS). In Figure 11b, the detection result is further segmented and refined as each segment based on (14) and (15), which are labeled with different colors. Finally, the BRFS is represented as the single BFRS in each group using the weighted clustering process based on (16). In Figure 11, it can be seen that the BFRS are dynamically generated based on the proposed BFRS detection method in Section 3.7. The proposed method primarily aims at the positioning of BFRS, but it can also represent the size of the obstacle. For the horizontal length of BFRS, it can be directly represented by the results of continuous detection, that is, without clustering the BFRS initially (as shown by the different BFRS group in Figure 11). The lateral width can be inferred from multi-source data coverage of the road surface, which helps estimate the lateral width of BFRS. The changes in z-axis acceleration from the recorded acceleration corresponding to detection results can indicate the vertical dimension (as shown by the actual acceleration recording in Figure 10).

In addition, from the acceleration data recorded, ideally, when a vehicle passes over a hump-type obstacle, it produces an upward acceleration followed by a downward acceleration. Conversely, passing over a hole results in the opposite pattern of acceleration. However, these features are not distinctly clear and are difficult to differentiate due to inherent noise in the acceleration data. Theoretically, since it uses spectrum analysis coefficients to represent BFRS features, and these coefficients are absolute values from the Fourier transform process, the directionality of accelerations caused by hump-type and hole-type obstacles is neutralized by the absolute value operation. Therefore, it is not possible to accurately capture the specific acceleration features of raised and lowered surface elements. What is more, regarding road surface irregularities, such as selected thickness and type of paint, it is possible to sample these characteristics, represent them using the spectrum features, and input them into a neural network model as part of the training set. The network is subsequently trained to learn the subtle variations and distinctions among these features, thereby enabling the detection of these types of BFRS.

*4.4. BFRS Detection Comparison with Different Methods*

To have an intuitive understanding of the performance of the proposed BFRS detection method, different experiments are conducted, and related comparisons are made with other methods. Firstly, it is the performance comparison between the original CNN model and the proposed BFRS detection method on the test dataset (Area A shown in Figure 7). The BFRS is detected using the original CNN model without the dataset augmented based on the movement feature analyzed in Sections 3.2 and 3.3. The experiment is reconducted based on the proposed method with the same parameter as that of the original CNN model, and the result is depicted in Figure 12.

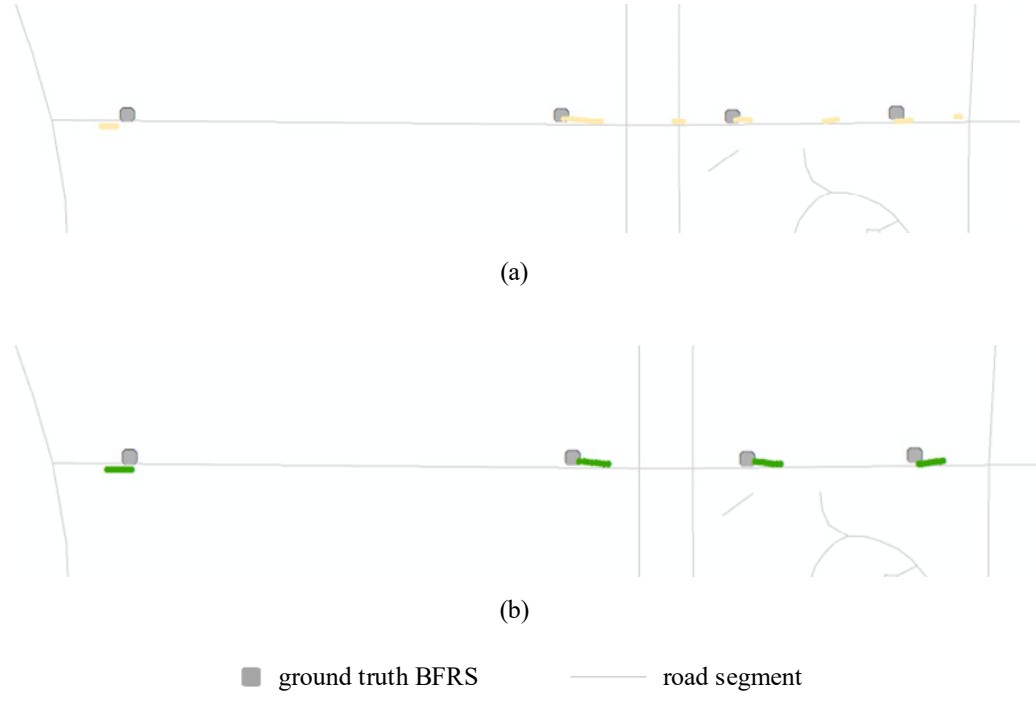

(a)

(b)

■ ground truth BFRS ——— road segment

○ BFRS result by original CNN • BFRS result by the proposed method

**Figure 12.** Comparison with and without augmentation information. (**a**) BRFS detected results without augmentation. (**b**) BRFS detected results with augmentation.

In the test dataset of Area A, there are four ground truth BFRS distributed along the road, as depicted by the gray square in Figure 12. In Figure 12a, it is the BFRS detection result based on the original CNN model [17] without augmentation information. Although four BFRS are covered by the detected BFRS groups, the results are over-detected, i.e., there

are seven BFRS groups detected and three are falsely detected. In comparison with the result in Figure 12a, the BFRS detection result in Figure 12b seems better; four BFRS are successfully covered by the detected BFRS group, and no more false-BFRS results are further detected. The over-detected BFRS result in Figure 12a is mainly caused by the insufficient training dataset, and it cannot detect BFRS in different movement situations. Hence, the proposed BFRS detection method turns out to be effective with augmentation information.

With the result in Figure 12, the BFRS detection results are further represented by the method in Section 3.6. Secondly, BFRS detection experiments are further conducted and compared with the adaptive threshold-based method [34] and the CWT-based method [7], as depicted in Figure 13.

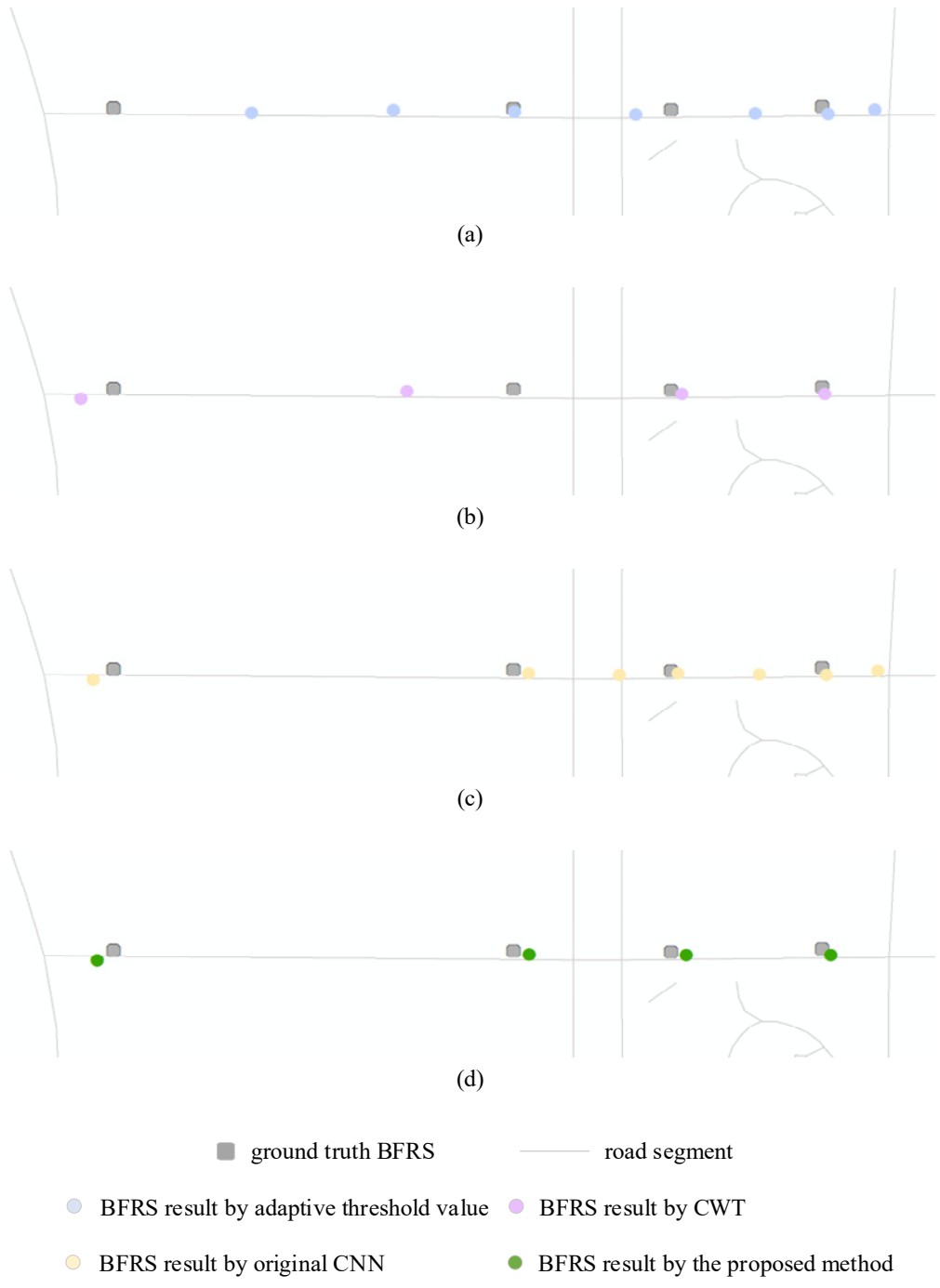

**Figure 13.** Comparisons between different BFRS detection methods. (**a**) Adaptive threshold value. (**b**) CWT. (**c**) Original CNN. (**d**) The proposed method.

In the result, the BFRS within the distance of 10 m is considered to be the correct BFRS, and the one out of the distance is taken as the wrong BFRS. In Figure 13a, it is the result by adaptive threshold value, computed based on the abnormal acceleration changes in its neighborhood. The first derivative of acceleration recordings is computed and taken as the abnormal change, along with the total standard deviation of the dataset. Then, the abnormal change larger than the standard deviation is adaptively selected as BFRS. Finally, seven BFRS are detected, with two correct BFRS and five wrong BFRS. In Figure 13b, the BFRS is computed based on the CWT method, with the parameters {wave name: 'db3', total scale: 256}, and the threshold value adaptively computed based on the absolute mean value of the coefficient matrix. The acceleration recording with a related coefficient larger than ten times the threshold value is selected as BFRS, and four BFRS are detected, with two correct BFRS and two wrong BFRS. Consistent with the experiment in Figure 12, the BFRS detection results of the original CNN and the proposed method are depicted in Figure 13c,d. In Figure 13c, seven BFRS are detected by the original CNN, with four correct BFRS and three wrong BFRS, while in Figure 13d, four BFRS are all correctly detected by the proposed method.

$$Precision = \frac{TP}{TP + FP} \tag{17}$$

$$Recall = \frac{TP}{TP + FN} \tag{18}$$

To evaluate the performances of different methods, precision and *recall* are computed based on (17) and (18). In the equation above, *TP* is the true positive BFRS (the true BFRS that successfully detected), *FP* is the false positive BFRS (the BFRS that falsely detected), and *FN* is the false negative BFRS (the true BFRS that are not detected). Finally, the *F-score* is further computed based on these two quality indexes, as denoted in (19), and the statistical result is depicted in Table 2.

$$F\text{-}score = 2 \times \frac{Precision \times Recall}{Precision + Recall} \tag{19}$$

**Table 2.** Statistic of the comparison results.

|  | Detected BFRS | Correct BFRS | Wrong BFRS | *Precision* | *Recall* | *F-Score* |
|---|---|---|---|---|---|---|
| Adaptive threshold value | 7 | 2 | 5 | 28.57% | 50.00% | 0.3636 |
| CWT | 4 | 2 | 2 | 50.00% | 50.00% | 0.5000 |
| Original CNN | 7 | 4 | 3 | 57.14% | 100.00% | 0.7272 |
| The proposed method | 4 | 4 | 0 | 100.00% | 100.00% | 1.0000 |

In Table 2, the *precision*, *recall*, and *F-score* are also computed, with the number of correct BFRS and wrong BFRS. For example, the number of detected BFRS, correct BFRS, and wrong BFRS by the adaptive threshold value are 7, 2, and 5, respectively, and the related *TP*, *FP*, and *FN* can be computed as 2, 5, and 2. Hence, the *precision* and *recall* are 28.51% and 50.00% based on (17) and (18), and the *F-score* is computed as 0.3636. The *precision*, *recall*, and *F-score* for the CWT method {*TP*: 2, *FP*: 2, *FN*: 2} are 50.00%, 50.00%, and 0.5000, respectively, and these indexes for the original CNN method {*TP*: 4, *FP*: 3, *FN*: 0} are 57.14%, 100.00%, and 0.7272. However, the *precision*, *recall*, *and F-score* for the proposed method {*TP*: 4, *FP*: 0, *FN*: 0} are 100.00%, 100.00%, and 1.0000, since all BFRS are correctly detected with no wrong result.

In the experiment result in Figure 13 and Table 2, more BFRS are detected than other methods by the adaptive threshold method, while most of them are wrong results since the threshold value cannot be adapted to every situation. It seems that the CWT method can perform well with four detected BFRS, while it also faces the similar situation of the adaptive threshold value method, and the threshold only fits limited conditions. In the original CNN method, the detection standard can be learned from the training dataset, with four corrected BFRS detected. However, the performance still depends on the training

dataset, and it performs not so well since there are three wrongly detected BFRS. Finally, four BFRS are successfully detected with the highest *F-score* by the proposed method with a movement–feature-augmented training dataset. As shown in the experiments, the proposed method achieved superior BFRS detection results, with the detected positions closely aligning with the ground truth BFRS locations. Unlike the CWT method and adaptive threshold method, the proposed method does not require the setting of specific parameters. Furthermore, compared to the original CNN method, the proposed method enhances the data by considering the characteristics of vehicle movements and multi-sensor recordings, resulting in superior experimental outcomes. Hence, it can be concluded that the BFRS detection result computed by the proposed method outperforms other results in the experiment. In addition, variations in tire or suspension systems across different vehicles may impact the BFRS detection results, and BFRS might not be adequately captured just by a single vehicle type. However, it assumes general conditions, and the credibility of BFRS detection results can be further improved through the integration of multi-source data.

### 4.5. BFRS Detection and Comparison in the Different Areas

To assess the capability of the proposed BFRS detection method, multi-sensor stream data in different areas are collected using smartphones, and experiments are conducted based on the trained deep learning model in the previous section. The sampling rate for the GPS sensor is 1 Hz, and that for the gyroscope and accelerometer is 100 Hz. The research area and discrete-sampled GPS stream data are depicted in Figure 14a,b, with augmentation information visualized by graded color.

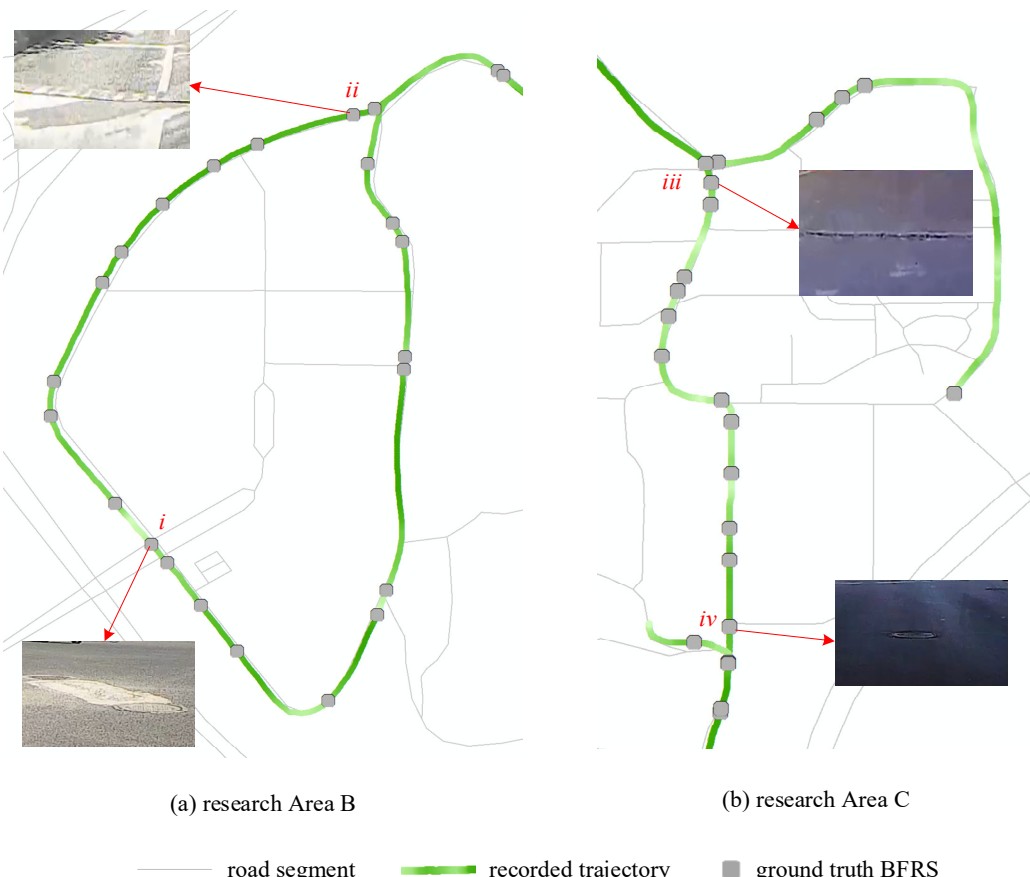

(a) research Area B                    (b) research Area C

road segment ——— recorded trajectory ▬▬▬ ground truth BFRS ◼

**Figure 14.** Different research areas. (**a**) Research Area B. (**b**) Research Area C.

In Figure 14, different kinds of BFRS in research Area B and C are visualized as the related pictures of (*i*) cracks, (*ii*) speed breaker, (*iii*) transverse gutter, and (*iv*) pothole. The detailed statistics of the collected multi-sensor stream data are depicted in Table 3.

**Table 3.** Statistic of collected multi-senor stream data in different areas.

|  | Length | Duration | GPS Records | Acceleration Records | Orientation Records |
|---|---|---|---|---|---|
| Area B | 1171 m | 234 s | 234 | 23,233 | 23,233 |
| Area C | 1475 m | 1298 s | 1298 | 129,081 | 129,081 |

With the test dataset in Areas B and C, BFRS are firstly collected using the proposed method, refined and segmented as different BFRS groups, and clustered and represented as BFRS objects. Then, comparisons are conducted using the adaptive threshold value method, CWT method, and original CNN method, using the same parameter as that in Area A. Comparisons with different BFRS detection methods in Area B are depicted in Figure 15.

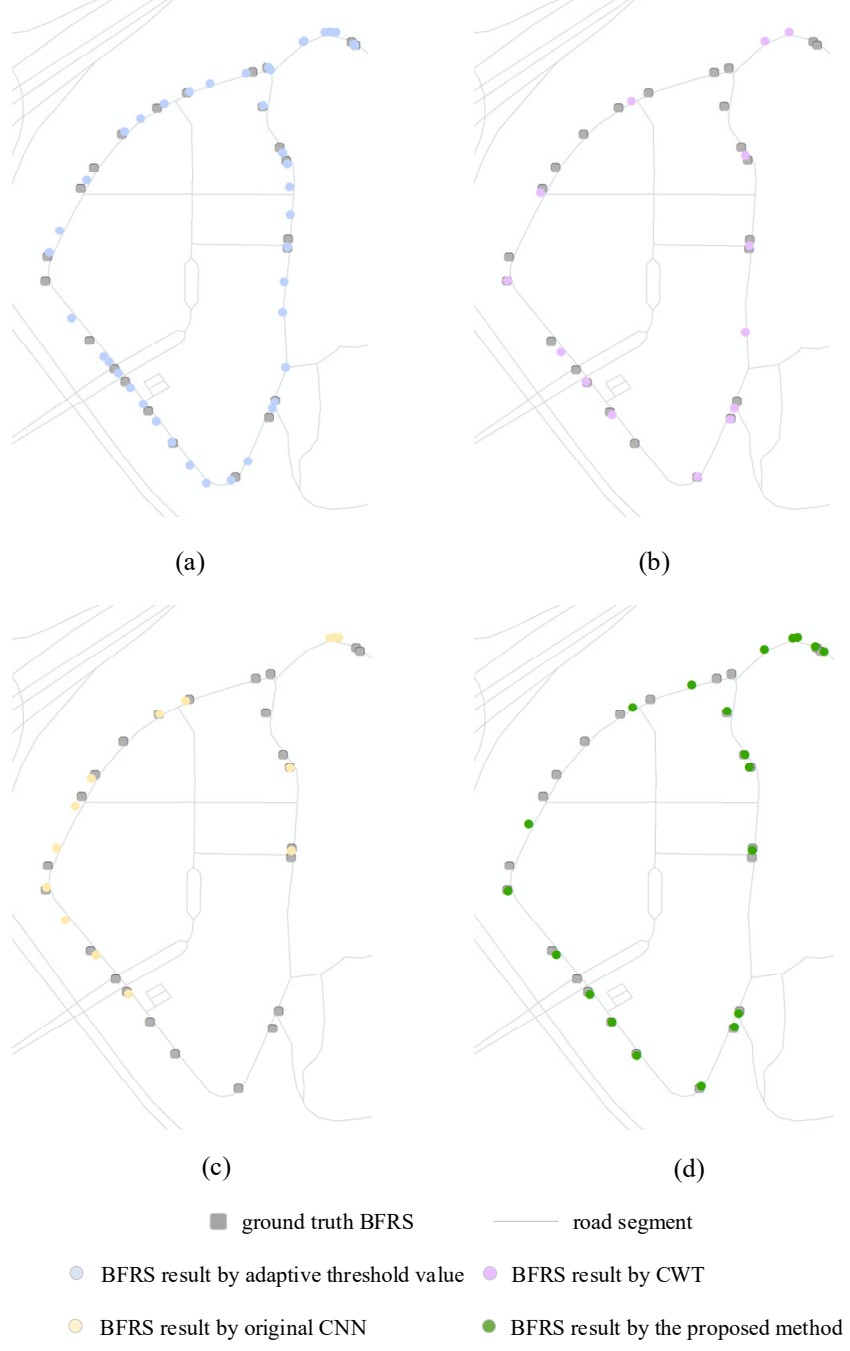

**Figure 15.** Comparisons with different BFRS detection methods in Area B. (**a**) Adaptive threshold value. (**b**) CWT. (**c**) Original CNN. (**d**) The proposed method.

There are 24 ground truth BFRS in Area B, while the number of detected BFRS varies in different results. In Figure 15a, there are 41 BFRS detected based on the adaptive threshold value method, and 18 BFRS are correctly detected. However, there are 23 wrong BFRS in the result, which holds the highest value compared with other methods. In Figure 15b,c, there are 14 BFRS detected in each result based on the CWT method and the original CNN method, while it performs a little better in the result of the CWT method with nine correct BFRS and five wrong BFRS. In contrast to the result of Figure 15c, the performance of the proposed method is dramatically improved, and 20 BFRS are detected, with 14 correct BFRS and 6 wrong BFRS, since the model is trained with the augmented dataset based on movement feature. The detailed statistics of the experiment results for Area B are depicted in Table 4.

**Table 4.** Statistics of the comparison results in Area B.

|  | Detected BFRS | Correct BFRS | Wrong BFRS | *Precision* | *Recall* | *F-Score* |
|---|---|---|---|---|---|---|
| Adaptive threshold value | 41 | 18 | 23 | 43.90% | 75.00% | 0.5538 |
| CWT | 14 | 9 | 5 | 64.29% | 37.50% | 0.4737 |
| Original CNN | 14 | 8 | 6 | 57.14% | 33.33% | 0.4210 |
| The proposed method | 20 | 14 | 6 | 70.00% | 58.33% | 0.6363 |

In Table 4, the *precision*, *recall*, and *F-score* are computed according to the experiment result in Figure 15. Although there are 18 correct BFRS detected by the adaptive threshold value method, the number of wrong BFRS is also the highest among all of the results, and the related *TP*, *FP, and FN* are computed as 18, 23, and 6. Hence, the *precision*, *recall*, and *F-score* are 43.90%, 75.00%, and 0.5538 for the first method. With the similar detection result by the CWT method and the original CNN method, the related *TP*, *FP*, and *FN* are computed as CWT: {*TP*: 9, *FP*: 5, *FN*: 15} and original CNN: {*TP*: 8, *FP*: 6, *FN*: 16}, respectively. Moreover, the *precision*, *recall*, and *F-score* are 64.29%, 37.50%, and 0.4737 for the CWT method, and 57.14%, 33.33%, and 0.4210 for the original method. Finally, the *precision*, *recall*, *and F-score* for the proposed method {*TP*: 14, *FP*: 6, *FN*: 10} are 70.00%, 58.33%, and 0.6363. With the highest *precision*, *recall*, and *F-score*, the proposed method performs best among all results.

As shown in Figure 15 and Table 4 of Area B, where BFRS are more prevalent, detecting all BFRS is challenging. However, the adaptive threshold method detected a higher number of BFRS but resulted in many false detections, leading to low precision. The CWT method applies continuous wavelet transformation to the acceleration data, resulting in complex wavelet coefficient matrixes that are difficult to distinguish using specific thresholds. The original CNN method, lacking training on a dataset enhanced based on movement characteristics, did not perform well in the new experimental area, detecting only 14 BFRS. In contrast, the method proposed in this article, which is based on the analysis and training of vehicle motion characteristics, identified 20 BFRS with a high *precision* and the best relative *F-score*, demonstrating the superior performance of our method.

To further evaluate the performance of the proposed detection method, the BFRS detection operation is conducted in Area C, along with the related comparisons with the adaptive threshold value method, CWT method, and original CNN method, and uses the same parameter as that in Area A. The result is depicted in Figure 16.

In Area C, there are 23 ground truth BFRS spatially distributed, as shown by the gray rectangle in Figure 16. In Figure 16a, 27 BFRS are detected by the adaptive threshold value method, with 14 correct BFRS and 13 wrong BFRS. In Figure 16b,c, the detected BFRS are 11 by the CWT method and 9 by the original CNN method, and both of them perform similarly with 4 wrong BFRS; however, there are more correct BFRS in Figure 16b than that of Figure 16c. Compared with other results, there are 22 BFRS detected by the proposed method, and 17 BFRS are correctly detected, which is the highest value among all results. The detailed statistics of the experiment results for Area C are depicted in Table 5.

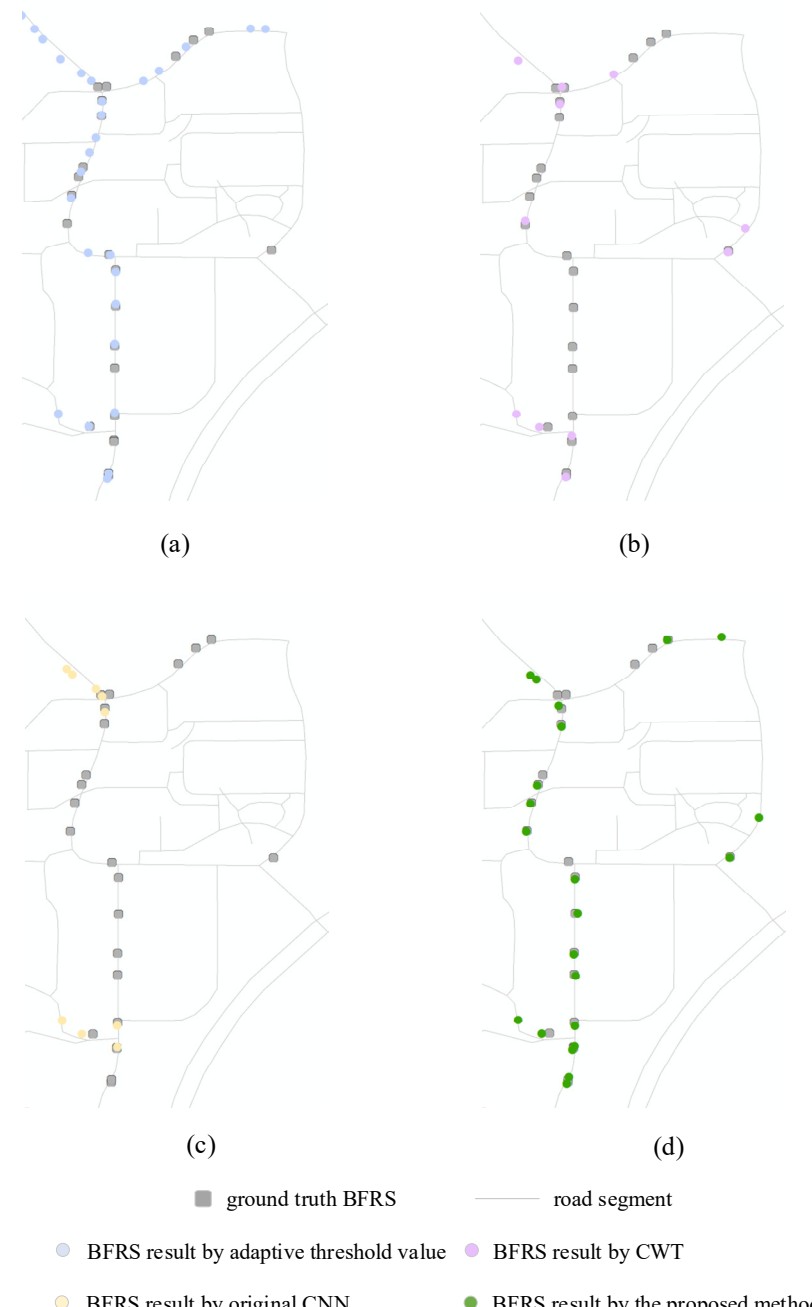

**Figure 16.** Comparisons with different BFRS detection methods in Area C. (**a**) Adaptive threshold value. (**b**) CWT. (**c**) Original CNN. (**d**) The proposed method.

**Table 5.** Statistics of the comparison results in Area C.

|  | Detected BFRS | Correct BFRS | Wrong BFRS | *Precision* | *Recall* | *F-Score* |
|---|---|---|---|---|---|---|
| Adaptive threshold value | 27 | 14 | 13 | 51.85% | 60.87% | 0.56 |
| CWT | 11 | 7 | 4 | 63.64% | 30.43% | 0.4117 |
| Original CNN | 9 | 5 | 4 | 55.56% | 21.74% | 0.3125 |
| The proposed method | 22 | 17 | 5 | 77.27% | 73.91% | 0.7555 |

In Table 5, the *precision*, *recall*, and *F-score* are computed according to the experiment result in Figure 16. With 14 correct BFRS and 13 wrong BFRS by the adaptive threshold value method, the related *TP, FP, and FN* are computed as 14, 13, and 9, and the *precision, recall,*

and *F-score* are 60.87%, 75.00%, and 0.5600 for the result in Figure 16a. With seven correct BFRS and four wrong BFRS in Figure 16b by the CWT method, the result in Figure 16b {*TP*: 7, *FP*: 4, *FN*: 16} outperforms that of Figure 16c by the original CNN method {*TP*: 5, *FP*: 4, *FN*: 18}. However, the *precision and recall* of the proposed method are 77.27% and 73.91%, according to the statistic results {*TP*: 15, *FP*: 7, *FN*: 6}; hence, the *F-score* is the highest among all results {adaptive threshold value: 0.5600; CWT: 0.4117, original CNN: 0.3125; the proposed method: 0.7555}. More correct BFRS and less wrong BFRS are detected based on the proposed method, and it performs best among all experimented results.

As shown in Figure 16 and Table 5 of Area C, where the spatial distribution of BFRS is denser compared to the experimental Area B, the adaptive threshold method detected a larger number of BFRS, but many were incorrect, resulting in low *precision*. The CWT method showed higher detection *precision*, yet selecting appropriate BFRS based on wavelet coefficients remains challenging. The original CNN detected 9 BFRS, whereas the proposed method detected 22 BFRS, with 17 being correct BFRS. Therefore, compared to these methods, the *F-score* performance of the proposed method is the best, consistent with the results from Area B.

With experiment results in Figures 15 and 16, Tables 4 and 5, BFRS can be detected with high *precision*, *recall*, and *F-score* based on the proposed method. Although there are more BFRS detected by the adaptive threshold value method, the *precision* is not good, as there are also too many wrong BFRS in the result. The CWT method performs better than the adaptive threshold value with a higher *precision*, but the number of detected BFRS is small, as is the result of the original CNN method. However, using an augmented dataset based on the movement feature and the learnable parameters, BFRS are detected with the highest quality index, such as *precision*, *recall*, and *F-score*. Finally, the proposed method performs best in the comparison. The experiments demonstrate that the proposed method described in this article, which establishes a feature representation for BFRS through spectrum modeling, enhances the BFRS training dataset using vehicle motion characteristics, and detects BFRS in different experimental areas using the CNN, achieves relatively superior experimental results compared to other methods. This indicates the applicability and superior performance of the proposed method.

## 5. Conclusions and Future Work

Road infrastructure is a critical component of the geographic data required for the development of smart cities. It plays a pivotal role in city transportation systems, where the integrity of road surfaces directly influences transportation safety and the quality of the driving experience. The detection and management of road surface irregularities, such as BFRS, are essential due to the stresses imposed by extensive vehicle use and infrastructure wear. To detect BFRS in an economical way, smartphones assembled with multiple sensors are applied in this study. By harnessing the ubiquity and sophisticated sensor capabilities of these devices, discrete-sampled and non-homogeneous datasets are collected. The relationship between BFRS and vehicle speed conditions was analyzed, leading to the development of a robust model that effectively identifies BFRS. Through preprocessing of three-axis acceleration stream data, a comprehensive BFRS model is constructed and augmented by the training dataset with movement features. Then, the deep learning neural network is designed, and the multi-dimensional spectrum feature is input into the neural network. Finally, BFRS are refined and weighted clustered according to detected points and the related probability. Multi-sensor stream data at different areas are collected, training datasets are augmented, spectrum features are constructed based on the modeling result, and the deep learning neural network is trained and validated. The proposed approach not only demonstrated superior performance in detecting BFRS compared to other existing methods, but also provided initial locations of the distress or damages of the road surface, which would contribute to the maintenance of road assets. The flexibility of the CNN model, which can be adapted or replaced with alternative architectures, adds to the scalability of the approach. The smartphone sensors used in this article are highly integrated within

the device. However, in IoT sensor application scenarios, especially when dealing with multiple sensors that lack necessary external enclosures and have lower levels of internal integration, calibrating sensors is essential. Additionally, the external environment (such as external magnetic fields) and the impact of different disturbances require in-depth study.

Future research aims to improve the performance of BFRS detection results with multi-modal sensors; for example, since video datasets can also be collected using smartphones, the deep learning neural network framework can be improved and redesigned to encode multi-modal stream data. Moreover, for vehicles with different types of tires and suspension systems, it is essential to explore the analysis of forces while overcoming an obstacle. This could provide deeper insights into the interaction between the vehicle's tires, suspension, and road surface, which could be crucial for encoding the BFRS and interpreting the results more accurately. Another aspect that needs to be further explored is BFRS detection with crowdsourced datasets, which collect road surface conditions using volunteered geographic information.

**Author Contributions:** H.L. was responsible for methodology, analysis, and drafting of the manuscript. Q.Z. contributed to the implementation, programming, and testing. D.J. was responsible for literature reviews, background knowledge, and improvement of the writing. J.H. contributed to the data collection and visualization of the results. All authors have read and agreed to the published version of the manuscript.

**Funding:** This research was funded by the National Natural Science Foundation of China [Grant No. 42101466].

**Institutional Review Board Statement:** Not applicable.

**Informed Consent Statement:** Not applicable.

**Data Availability Statement:** The data that were analyzed in this study can be found at https://doi.org/10.6084/m9.figshare.26039233.v1. The street map is available from OpenStreetMap, accessed on 19 March 2024.

**Acknowledgments:** We would like to give special thanks to Dieter Pfoser at George Mason University for their comments and revisions that have improved the quality of this article.

**Conflicts of Interest:** The authors declare no conflicts of interest.

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
