# Peer review of "Bump Feature Detection Based on Spectrum Modeling of Discrete-Sampled, Non-Homogeneous Multi-Sensor Stream Data"

_applsci, doi:10.3390/app14156744_

Round 1
Reviewer 1 Report (Previous Reviewer 1)
Comments and Suggestions for Authors
The work improved significantly since the first submission.
I have a single comment concerning the following sentence in the conclusion:
“The proposed approach not only demonstrated superior performance in detecting BFRS compared to other existing methods but also contributed to the maintenance and optimization of road assets.”
How did the approach contribute to the maintenance and optimization of road assets?
The authors presented a method to detect bumps. This is only the first step in the complex process of road maintenance.
Author Response
The work improved significantly since the first submission.
Response: Thanks for your valuable feedback. Your insights have significantly enhanced the quality of our paper.
I have a single comment concerning the following sentence in the conclusion:
“The proposed approach not only demonstrated superior performance in detecting BFRS compared to other existing methods but also contributed to the maintenance and optimization of road assets.”
How did the approach contribute to the maintenance and optimization of road assets?
The authors presented a method to detect bumps. This is only the first step in the complex process of road maintenance.
Response: Completely agreed. The method proposed in this manuscript primarily focuses on detecting bump features on road surfaces, and it’s just the very beginning of the comprehensive road maintenance process. As to the road maintenance, extensive works (such as the accurate surveying and road repairing) are needed after the bump feature detection of road surfaces, and the method just serves to get the initial locations of distresses or damages on the road surface. Recognizing the broader context, we have revised the relevant descriptions in the Conclusion section to better reflect this.
“The proposed approach not only demonstrated superior performance in detecting BFRS compared to other existing methods but also provided initial locations of distresses or damages of the road surface, which would contributed to the maintenance of road assets.”
Thank you very much for your insightful comments.
Reviewer 2 Report (New Reviewer)
Comments and Suggestions for Authors
The authors undertook an interesting analysis of road surface irregularities using tools dedicated to mobile phones. Sensors used in phones have, of course, limited accuracy, but they are used in a wide range of applications.
The entire work is written correctly, but it lacks important practical information related to the measurement process itself, the correlation of measurements from various sources and the accuracy of integrating these measurements.
By making measurements using acceleration sensors and gyroscopes, you can easily determine the vehicle's speed and the distance traveled using numerical integration. To do this correctly, you need to rotate the regional coordinate system related to the vehicle to the global coordinate system related to the road. The lack of such settings correction generates an apparent increase in acceleration values ​​while the vehicle is moving.
How was this rotation of the coordinate systems accomplished?
What mathematical method was used?
The acceleration graphs shown look as if they were filtered. Acceleration measurements with a frequency of 100 Hz have a lot of noise. This noise is most often distributed symmetrically around the expected value and in order to present a visually friendly graph, appropriate filtering methods of the measurement signal are used. Unfortunately, most of them introduce interference in the form of a time shift of the signal.
How is it in this case?
Is there a digital filter in the measuring system?
If so, what kind and what delay does the signal introduce?
How was the signal from the 100Hz sensors correlated with the 1Hz GPS signal?
Were base points used in the path analysis?
/Very useful for analyzing accelerations, speeds and distances determined from acceleration signals/
Detecting surface unevenness using a camera and microphone is fully justified, because even small unevenness usually generates sound. Similarly to the so-called playing road: Hungary road 67 - playing Road. There are some irregularities here, but they are not damage to the surface. However, it can be noticed that the selected thickness and type of paint causes a change in the noise of the tire-surface interaction.
When using acceleration sensors, remember that unevenness is absorbed by both the tire and the vehicle's suspension. In my experience, low profile bumps or long period bumps do not show up in the acceleration results while the vehicle is moving. However, the impact of the tire on an obstacle is visible. When approaching an obstacle, the graphs clearly show "impulses" in the form of momentary high acceleration values ​​in both the Z and X axes. However, when comparing different cars while overcoming an obstacle of a defined size, differences resulting from different suspension systems are clearly visible. The above-mentioned experimental studies also showed that the vehicle speed has a smaller impact on the value of instantaneous acceleration. This is caused by greater compliance of the tire and suspension at higher speeds, which can be easily observed, for example, during car races on gravel surfaces, where the suspension and tire float are visible and the vehicle body moves stably.
What are the smallest inequalities that can be observed using acceleration sensors?
What values ​​of instantaneous accelerations when hitting an obstacle occur in the Z axis and in the X axis?
The use of gyroscopic sensors made of piezoelectric materials has a serious disadvantage. These sensors tend to have signal leakage over longer measurement times.
How was the influence of signal flow in gyroscopes eliminated?
There is no clear statement of the purpose of the work and no indication of the evaluation criteria.
How to distinguish a hump-type obstacle from a hole in the surface?
What are the smallest obstacles we can detect with the proposed method?
Can the size of the obstacle be determined based on the measurement results?
/Please note that we do not know the tire deflection and suspension/
The last paragraph of the introduction is not necessary, it is a research paper, not a diploma thesis.
Author Response
The authors undertook an interesting analysis of road surface irregularities using tools dedicated to mobile phones. Sensors used in phones have, of course, limited accuracy, but they are used in a wide range of applications.
Response: It’s so nice to receive your professional comments, and we have revised the manuscript to improve the quality of it.
The entire work is written correctly, but it lacks important practical information related to the measurement process itself, the correlation of measurements from various sources and the accuracy of integrating these measurements.
Response: Thanks for the insightful comments.
In our study, we conducted experiments using acceleration, orientation, and GPS data acquired from smartphones' built-in motion sensors. We constructed spectrum features from the acceleration data and fed them into a neural network to detect bump features of the road surface (BFRS). Due to inherent noise, we did not rely on direct numerical changes in acceleration to measure BFRS.
Furthermore, our approach models the movement of vehicles and analyzes the relationship between speed and BFRS characteristics, rather than focusing on data from multiple smartphones or varying vehicle types. Consequently, the manuscript does not extensively describe the data acquisition and measurement process across different sensors or vehicles.
Compared to using professional surveying equipment for BFRS measurement, our method leverages smartphones to enhance the efficiency of BFRS detection. It provides initial positioning information for road maintenance and preliminary location data for subsequent construction projects.
Our method clusters detected BFRS points, using the cluster centers to represent BFRS locations and compares these with actual BFRS positions. We consider detections within 10 meters of the actual BFRS as accurate, using these metrics to calculate precision, recall, and F1-score, and to evaluate the performance of different BFRS detection methods.
We have made explicit explanations and revised the related descriptions in the manuscript.
“To address these challenges and detect BFRS using the widely used motion sensor, this article concentrates on multi-sensor stream data obtained from smartphones positioned in arbitrary poses,”
“Rather than the numerical measurement of acceleration data from multiple smartphones or varying vehicle types, the contributions of this article are (1) the modeling of vehicle movement than can deal with the BFRS in different frequency and driving condition, along with the augmentation of the training dataset; (2) the representing and detecting of BFRS in account with the noisy sampling; (3) the presentation and integration of BFRS locations based on the weighted clustering method.”
“Different with the traditional IoT sensor, it does not extensively describe the data acquisition and measurement process across different sensors or vehicles. Instead, it focuses on acceleration, orientation, and GPS data acquired from smartphones' built-in motion sensors.”
“Subsequently, spectral analysis techniques [38] are employed to represent the spectral features of the acceleration sampling data, which doesn’t rely on direct numerical changes in acceleration to measure BFRS due to inherent sensor noise.”
“The proposed approach not only demonstrated superior performance in detecting BFRS compared to other existing methods but also provided initial locations of distresses or damages of the road surface, which would contributed to the maintenance of road assets.”
“After that operation, the cluster result is compared with the actual BFRS positions, treating detections within 10 meters of the actual BFRS as correct, and those outside this range as wrong. These criteria serve as our evaluation metrics, and precision, recall, and F1-score are calculated. Based on the criteria, comparisons are conducted with different BFRS detection methods, in addition, BFRS in different areas are computed and results are evaluated with different methods.”
By making measurements using acceleration sensors and gyroscopes, you can easily determine the vehicle's speed and the distance traveled using numerical integration. To do this correctly, you need to rotate the regional coordinate system related to the vehicle to the global coordinate system related to the road. The lack of such settings correction generates an apparent increase in acceleration values ​​while the vehicle is moving.
Response: Completely agreed.
Indeed, our methodology also employs spatial transformation techniques to convert vehicle coordinates (based on results recorded by smartphone sensors) to regional coordinates.
In our approach, a smartphone arbitrary placed within the vehicle serves as the data source. Using the smartphone's recorded 3D orientation information, we perform spatial transformations on the corresponding acceleration data at each moment. This converts the arbitrarily oriented 3D acceleration data to acceleration information parallel to the XY plane and perpendicular to the Z-axis of the road surface. This process accurately represents the vertical acceleration relative to the road surface and the horizontal changes in vehicle movement at each moment.
We have made explicit explanations and revised the related descriptions in the manuscript.
“Using the smartphone's recorded 3D orientation information, spatial transformations are performed on the corresponding acceleration data at each moment. This process converts the arbitrarily oriented 3D acceleration data into acceleration information that is parallel to the xy plane and perpendicular to the z-axis of the road surface, thereby accurately representing the vertical acceleration relative to the road surface and the horizontal changes in vehicle movement at each moment.”
How was this rotation of the coordinate systems accomplished?
What mathematical method was used?
Response: Accepted.
As the above two issues pertain to the same concern, we will address them together. Our study implemented a coordinate system rotation using the Direction Cosine Matrix (DCM) method. At each sampling moment of acceleration, we utilized linear interpolation to temporally align sensor attitude data. Using the three-dimensional posture data corresponding to that moment, we established a DCM matrix. This matrix was then multiplied with the corresponding tri-axial acceleration data to achieve spatial transformation, yielding acceleration data with the Z-axis perpendicular to the road surface and the XY-axis parallel to it.
In professional setups, sensors installed at fixed positions and orientations can accurately record vehicle states, enabling the capture of the vehicle's instantaneous direction of travel. This allows alignment of the vehicle's directional axis with the acceleration x-axis. However, in our method, the placement of smartphones is more flexible, and aligning the sensor's x-axis with the vehicle's direction of travel requires professional calibration. The BFRS detection process is intended for more general, non-professional scenarios. When a vehicle passes over a BFRS, noticeable vibrations occur in the Z-direction, and the vehicle's speed in the XY direction is somewhat reduced. The detection of BFRS is more closely related to the Z-axis and the direction of vehicle travel. Therefore, our paper does not strictly align the acceleration in the XY direction with the direction of vehicle travel, and the x-direction is not necessarily the direction of vehicle travel.
We have made explicit explanations and revised the related descriptions in the manuscript.
“The proposed method implements the coordinate system rotation using the Direction Cosine Matrix (DCM) method. At each sampling moment of acceleration, linear interpolation is utilized to temporally align sensor attitude data. Using the three-dimensional posture data corresponding to that moment, a DCM matrix is established. This matrix is then multiplied with the corresponding tri-axial acceleration data to achieve spatial transformation, yielding acceleration data with the z-axis perpendicular to the road surface and the xy-axis parallel to it.”
“Additionally, it can be observed that BFRS impacted both the vertical (z-axis) and horizontal (xy-axis) directions of the vehicle. In the z-axis direction, the changes in acceleration are quite noticeable, typically fluctuating above ±3m/s². In the x-axis direction (assuming it is the direction of vehicle travel), a noticeable deceleration effect is also presented, although the changes in acceleration were relatively minor.”
The acceleration graphs shown look as if they were filtered. Acceleration measurements with a frequency of 100 Hz have a lot of noise. This noise is most often distributed symmetrically around the expected value and in order to present a visually friendly graph, appropriate filtering methods of the measurement signal are used. Unfortunately, most of them introduce interference in the form of a time shift of the signal.
Response: Thanks for your professional comments.
In our study, we used the MATLAB Mobile app to record data from smartphone sensors, and did not apply any filtering to the data. Thanks to the BFRS spectrum representation method used in our research, this approach exhibits a certain tolerance to noise. It transforms one-dimensional acceleration changes into two-dimensional spectrum features. By distinguishing BFRS characteristics from noise across different frequency spectrum coefficients, this method effectively reduces the impact of noise on experimental results. We use a neural network to learn and detect these BFRS characteristics, marking a clear distinction from methods that rely directly on acceleration thresholds.
We have made explicit explanations and revised the related descriptions in the manuscript.
“Smartphones served as the direct source for sensor data collection, with raw sensor data being recorded using the MATLAB Mobile app,”
“No data filtering methods were employed, including for acceleration data.”
“which doesn’t rely on direct numerical changes in acceleration to measure BFRS due to inherent sensor noise.”
“Despite the presence of noise in the original recordings, the method transforms one-dimensional acceleration changes into two-dimensional spectral features.”
How is it in this case?
Is there a digital filter in the measuring system?
If so, what kind and what delay does the signal introduce?
Response: Accepted.
As the above three issues pertain to the same concern, we will address them collectively. In our study, smartphones were used as the direct source of sensor data, with raw sensor data recorded using the MATLAB Mobile app. We did not employ any data filtering methods, including for acceleration data. Our approach utilizes spectrum feature representation to characterize BFRS features, which is capable of handling noise situations effectively.
We have made explicit explanations and revised the related descriptions in the manuscript.
“with raw sensor data being recorded using the MATLAB Mobile app”
“No data filtering methods were employed, including for acceleration data”
“By distinguishing BFRS characteristics from noise across different frequency spectrum coefficients, the proposed method effectively reduces the impact of noise on experimental results.”
How was the signal from the 100Hz sensors correlated with the 1Hz GPS signal?
Were base points used in the path analysis?
/Very useful for analyzing accelerations, speeds and distances determined from acceleration signals/
Response: Accepted.
As the above two issues relate to the same concern, we will address them together. In our method, we utilized smartphones to gather accelerometer and orientation data at 100 Hz and GPS data at 1 Hz. During data preprocessing, to correlate acceleration sampling moments with GPS location data, we employed linear interpolation to estimate GPS positions at each acceleration sampling moment.
We assume that under normal driving conditions (speeds of 0-60 km/h, without sudden traffic accidents, and consistent movement), the changes in speed and position over a 1-second interval are continuous. Additionally, considering that GPS signals inherently have an error even greater than 5 meter, and our evaluation criteria use a 10-meter buffer zone around the actual BFRS to distinguish between correct and incorrect detection results, it is reasonable to use linear interpolation for GPS positions.
We have made explicit explanations and revised the related descriptions in the manuscript.
“It is assumed that under normal driving conditions (speeds of 0-60 km/h, without sudden traffic accidents, and consistent movement), changes in speed and position over a 1-second interval are continuous. Furthermore, given that GPS signals inherently possess an error greater than 5 meters, employing linear interpolation for estimating GPS positions is considered reasonable. Consequently, to correlate acceleration sampling moments with GPS location data, linear interpolation is used to estimate GPS positions at each acceleration sampling moment.”
Detecting surface unevenness using a camera and microphone is fully justified, because even small unevenness usually generates sound. Similarly to the so-called playing road: Hungary road 67 - playing Road. There are some irregularities here, but they are not damage to the surface. However, it can be noticed that the selected thickness and type of paint causes a change in the noise of the tire-surface interaction.
Response: Completely agreed.
Using smartphone sensor data, particularly acceleration data for BFRS detection, is akin to "listening" to road surface changes through an accelerometer, similar to how a song records information about the road surface. However, unlike voice signals where a speaker’s speed is relatively stable, vehicle speed and acceleration vary in real-time based on actual road conditions. This variation can cause the same signal to have multiple expressions. Therefore, our paper focuses on analyzing and modeling the relationship between BFRS characteristics and vehicle speed, and on enhancing features based on this relationship to capture BFRS information under various speed conditions. This is one of the innovative aspects of our study.
Regarding road surface irregularities, such as the selected thickness and type of paint, we can sample these characteristics, represent them using our method, and input them into a neural network as part of the training set. The network is then trained to learn the subtle variations and distinctions among these features, thus enabling the detection of specific types of BFRS.
We have made explicit explanations and revised the related descriptions in the manuscript.
“Using smartphone sensor data, particularly acceleration data for BFRS detection, is akin to "listening" to road surface changes through an accelerometer, much like how a song records information about its environment. However, unlike voice signals where a speaker’s speed remains relatively stable, vehicle speed and acceleration fluctuate in real-time according to actual road conditions. This variability can cause the same signal to manifest in multiple ways. Consequently, one of the innovations of the proposed method is on analyzing and modeling the relationship between BFRS characteristics and vehicle speed, and on enhancing features based on this relationship to accurately capture BFRS information under various speed conditions.”
“What’s more, regarding road surface irregularities, such as selected thickness and type of paint, it is possible to sample these characteristics, represent them using the spectrum features, and input them into a neural network model as part of the training set. The network is subsequently trained to learn the subtle variations and distinctions among these features, thereby enabling the detection of these types of BFRS.”
When using acceleration sensors, remember that unevenness is absorbed by both the tire and the vehicle's suspension. In my experience, low profile bumps or long period bumps do not show up in the acceleration results while the vehicle is moving. However, the impact of the tire on an obstacle is visible. When approaching an obstacle, the graphs clearly show "impulses" in the form of momentary high acceleration values ​​in both the Z and X axes. However, when comparing different cars while overcoming an obstacle of a defined size, differences resulting from different suspension systems are clearly visible. The above-mentioned experimental studies also showed that the vehicle speed has a smaller impact on the value of instantaneous acceleration. This is caused by greater compliance of the tire and suspension at higher speeds, which can be easily observed, for example, during car races on gravel surfaces, where the suspension and tire float are visible and the vehicle body moves stably.
Response: Thank you for your professional comments, from which we have learned much.
Different vehicle models may exhibit varying responses to the same bump feature of the road surface due to differences in their suspension and tire systems. Our method infers road conditions from the sensor and trajectory data collected, focusing primarily on determining BFRS, and has not extensively studied the impact of the tire and vehicle's suspension systems.
Additionally, rather than comparing different vehicle types, our method used a more general vehicle (Honda CRV) for experiments, focusing more on the road surface BFRS information. If a single vehicle type does not capture the BFRS adequately, our method could be applied to crowd-sourced data, gathering road information from multi-source data across various vehicles. This remains an open question for further research.
We have made explicit explanations and revised the related descriptions in the manuscript.
“Through these operations, the CNN model can “hear” the voice of the road surface, and detect BFRS from the input dataset.”
“In addition, variations in tire or suspension systems across different vehicles may impact the BFRS detection results, and BFRS might not be adequately captured just by a single vehicle type. However, it assumes general conditions and the credibility of BFRS detection results can be further improved through the integration of multi-source data.”
“concentrates on multi-sensor stream data obtained from smartphones positioned in arbitrary poses”
“which doesn’t rely on direct numerical changes in acceleration to measure BFRS due to inherent sensor noise.”
What are the smallest inequalities that can be observed using acceleration sensors?
Response: Accepted.
Our research focuses on road surface BFRS, which exhibit distinct bump characteristics. We have not specifically collected training data for surface irregularities such as selected thickness and type of paint. However, we believe that these features could be detected if data on them were collected and used as input to our method.
Furthermore, we conducted a detailed statistical analysis of BFRS in our experimental area. Our field survey recorded variations in elevation ranging from 2.6 cm to 5 cm and horizontal lengths along the road from 3.8 cm to 72 cm. To describe these features more accurately, we have updated the definition of BFRS in our manuscript as follows: BFRS refers to surface features that have variations in elevation greater than 3 cm and extend horizontally along the road for more than 4 cm.
We have made explicit explanations and revised the related descriptions in the manuscript.
“In the research area, BFRS refers to surface features that have variations in elevation greater than 3 cm and extend horizontally along the road for more than 4 cm.”
“regarding road surface irregularities, such as selected thickness and type of paint, it is possible to sample these characteristics, represent them using the spectrum features, and input them into a neural network model as part of the training set. The network is subsequently trained to learn the subtle variations and distinctions among these features, thereby enabling the detection of these types of BFRS.”
What values of instantaneous accelerations when hitting an obstacle occur in the Z axis and in the X axis?
Response: Accepted.
We would like to address this question from two perspectives: (1) In our experiments, we observed that BFRS impacted both the vertical (Z-axis) and horizontal (XY-axis) directions of the vehicle. In the Z-axis direction, the changes in acceleration were quite noticeable, typically fluctuating above ±3m/s². In the X-axis direction (assuming it is the direction of vehicle travel), there was a noticeable deceleration effect, although the changes in acceleration were relatively minor. (2) However, using a threshold method to determine BFRS can lead to significant errors. Therefore, our study employs spectrum features to identify road surface BFRS, which also helps to some extent in mitigating the impact of acceleration noise on the experimental results.
We have made explicit explanations and revised the related descriptions in the manuscript.
“Additionally, it can be observed that BFRS impacted both the vertical (z-axis) and horizontal (xy-axis) directions of the vehicle. In the z-axis direction, the changes in acceleration are quite noticeable, typically fluctuating above ±3m/s². In the x-axis direction (assuming it is the direction of vehicle travel), a noticeable deceleration effect is also presented, although the changes in acceleration were relatively minor. However, employing a threshold method to determine BFRS can lead to significant errors due to the inherent noise in acceleration recordings. Therefore, the proposed method utilizes spectral features to detect BFRS, which also aids in mitigating the impact of acceleration noise to some extent.”
“By distinguishing BFRS characteristics from noise across different frequency spectrum coefficients, the proposed method effectively reduces the impact of noise on experimental results.”
“which doesn’t rely on direct numerical changes in acceleration to measure BFRS due to inherent sensor noise.”
The use of gyroscopic sensors made of piezoelectric materials has a serious disadvantage. These sensors tend to have signal leakage over longer measurement times.
How was the influence of signal flow in gyroscopes eliminated?
Response: Agreed.
Instead of using IoT sensor hardware to directly obtain data, our study collects data using sensors installed on smartphones. These devices benefit from built-in sensor calibration features within the operating system. For example, in iOS, settings can be adjusted via “System” -> “Privacy” -> “Location Services” -> “System Services” -> “Compass Calibration” and “Motion Calibration & Distance.” Additionally, we assume that the sensors are preliminarily calibrated during the hardware integration in smartphones. Before starting data collection, we restart the smartphone and leave it idle for a while to ensure that the device’s sensor calibration features are activated.
During data collection, our sampling duration generally lasts no longer than 10 minutes, which limits the accumulation of errors. The data is preprocessed using the smartphone's sensor calibration features before analysis. When detecting BFRS, considering potential time deviations in the detection results and the accuracy error in GPS positioning, we use a 10-meter search range as our criterion for comparison. Our experimental results have shown good consistency with actual conditions.
Since our focus is on analyzing the accelerometer data, we have not conducted an in-depth study of the specific hardware structure, particularly the influence of signal flow in gyroscopes.
We have made explicit explanations and revised the related descriptions in the manuscript.
“At the system level, calibrations can be made through the smartphone’s system settings to calibrate between multiple sensors (for example, in iOS, settings can be conducted via “System”->”Privacy”->”Location Services”->”System Services”->“Compass Calibration” and “Motion Calibration & Distance”). Before initiating data collection, the smartphone is restarted and left idle for a period to ensure activation of the device’s sensor calibration features.”
“During data collection, the sampling frequency of the smartphone's accelerometer and orientation sensors is consistent, and the impact of the sensors operating at short intervals, such as 100Hz, on the results is limited compared with vehicle speeds ranging from 0 km/h to 60 km/h.”
“and each collected dataset generally lasts no longer than 10 minutes.”
“After that operation, the cluster result is compared with the actual BFRS positions, treating detections within 10 meters of the actual BFRS as correct,”
“this article concentrates on multi-sensor stream data obtained from smartphones positioned in arbitrary poses”
There is no clear statement of the purpose of the work and no indication of the evaluation criteria.
Response: Accepted.
The objective of our work is to detect road surface BFRS using stream data acquired from smartphone sensors, and to evaluate the performance of our method based on the accuracy of the detected positions relative to their actual locations.
In assessing the experimental results, we take into consideration potential time deviations in BFRS detection and the accuracy error of GPS positions. As previously mentioned, our criterion for evaluation uses a 10-meter search range for comparisons. We cluster the detected BFRS trajectory points, using the cluster centers to represent BFRS locations. We compare these to the actual BFRS positions, treating detections within 10 meters of the actual BFRS as accurate, and those outside this range as inaccurate. These criteria serve as our evaluation metrics, and we use them to calculate precision, recall, and F1-score, comparing the experimental outcomes of different BFRS detection methods.
We have made explicit explanations and revised the related descriptions in the manuscript.
“To address these challenges and detect BFRS using the widely used motion sensor, this article concentrates on multi-sensor stream data obtained from smartphones positioned in arbitrary poses,”
“Rather than the numerical measurement of acceleration data from multiple smartphones or varying vehicle types, the contributions of this article are (1) the modeling of vehicle movement than can deal with the BFRS in different frequency and driving condition, along with the augmentation of the training dataset; (2) the representing and detecting of BFRS in account with the noisy sampling; (3) the presentation and integration of BFRS locations based on the weighted clustering method.”
“After that operation, the cluster result is compared with the actual BFRS positions, treating detections within 10 meters of the actual BFRS as correct, and those outside this range as wrong. These criteria serve as our evaluation metrics, and precision, recall, and F1-score are calculated. Based on the criteria, comparisons are conducted with different BFRS detection methods, in addition, BFRS in different areas are computed and results are evaluated with different methods.”
“In the result, the BFRS within the distance of 10 m is considered to be the correct BFRS and the one out of the distance is taken as wrong BFRS.”
How to distinguish a hump-type obstacle from a hole in the surface?
Response: Accepted.
From the acceleration data recorded, ideally, when a vehicle passes over a hump-type obstacle, it produces an upward acceleration followed by a downward acceleration. Conversely, passing over a hole results in the opposite pattern of acceleration. However, in practical experiments, we have found that due to inherent noise in the acceleration data, these features are not distinctly clear and are difficult to differentiate.
Theoretically, since our study uses spectrum analysis coefficients to represent BFRS features, and these coefficients are absolute values from the Fourier transform process, the directionality of accelerations caused by hump-type and hole-type obstacles is neutralized by the absolute value operation. Therefore, it is not possible to accurately capture the specific acceleration features of raised and lowered surface elements.
From a road maintenance perspective, identifying the initial location of road surface BFRS using our method is just the first step in road maintenance. The objective of providing auxiliary support for road maintenance through our approach has been achieved.
We have made explicit explanations and revised the related descriptions in the manuscript.
“In addition, from the acceleration data recorded, ideally, when a vehicle passes over a hump-type obstacle, it produces an upward acceleration followed by a downward acceleration. Conversely, passing over a hole results in the opposite pattern of acceleration. However, these features are not distinctly clear and are difficult to differentiate due to inherent noise in the acceleration data. Theoretically, since it uses spectrum analysis coefficients to represent BFRS features, and these coefficients are absolute values from the Fourier transform process, the directionality of accelerations caused by hump-type and hole-type obstacles is neutralized by the absolute value operation. Therefore, it is not possible to accurately capture the specific acceleration features of raised and lowered surface elements.”
“improve the efficiency of BFRS detection operations and combine user convenience with extensive data coverage”
“and detect BFRS using the widely used motion sensor”
“provided initial locations of the distress or damages of the road surface, which would contributed to the maintenance of road assets”
What are the smallest obstacles we can detect with the proposed method?
Response: Accepted.
As mentioned in a previous response, we conducted a field survey in the research area and identified variations in elevation ranging from 2.6 cm to 5 cm, and horizontal lengths along the road from 3.8 cm to 72 cm.
To more accurately describe these features, we have updated the definition in the article as follows: BFRS refers to surface features that have variations in elevation greater than 3 cm and extend horizontally along the road for more than 4 cm.
We have made explicit explanations and revised the related descriptions in the manuscript.
“In the research area, BFRS refers to surface features that have variations in elevation greater than 3 cm and extend horizontally along the road for more than 4 cm.”
Can the size of the obstacle be determined based on the measurement results?
/Please note that we do not know the tire deflection and suspension/
Response: Accepted.
Our method primarily aims at the positioning of BFRS, but it can also represent the size of the obstacle. For the horizontal length of BFRS, it can be directly represented by the results of continuous detection, that is, without clustering the BFRS initially. The lateral width can be inferred from multi-source data coverage of the road surface, which helps estimate the lateral width of BFRS. The changes in z-axis acceleration from the trajectory data corresponding to detection results can indicate the vertical dimension.
Regarding tire deflection and suspension, these are significant factors worth further investigation, especially in the context of the growing interest in autonomous driving. Since our method analyzes trajectory data recorded by smartphones directly, variations in tire or suspension systems across different vehicles may impact the BFRS detection outcomes. Relying solely on a single data source for BFRS inference could compromise the reliability of the results. However, our method assumes general conditions and improves the credibility of BFRS detection results through the integration of multi-source data.
Our approach represents just the initial step in road maintenance, marking the beginning of this work. Professional personnel and equipment are necessary for complete road maintenance, and our study only addresses the initial stages.
We have made explicit explanations and revised the related descriptions in the manuscript.
“The proposed method primarily aims at the positioning of BFRS, but it can also represent the size of the obstacle. For the horizontal length of BFRS, it can be directly represented by the results of continuous detection, that is, without clustering the BFRS initially (as shown by the different BFRS group in Figure 11). The lateral width can be inferred from multi-source data coverage of the road surface, which helps estimate the lateral width of BFRS. The changes in z-axis acceleration from the recorded acceleration corresponding to detection results can indicate the vertical dimension (as shown by the actual acceleration recording in Figure 10).”
“In addition, variations in tire or suspension systems across different vehicles may impact the BFRS detection results, and BFRS might not be adequately captured just by a single vehicle type. However, it assumes general conditions and the credibility of BFRS detection results can be further improved through the integration of multi-source data.”
The last paragraph of the introduction is not necessary, it is a research paper, not a diploma thesis.
Response: Accepted. The last paragraph of the introduction has been discarded from the manuscript.
Reviewer 3 Report (New Reviewer)
Comments and Suggestions for Authors
Title: The title of the article clearly defines the subject of the article.
Abstract: The abstract clearly indicates the goal and subject of research based on the Bump Features of the Road Surface. However, a proposal for improving the work refers to adding information about the data sample as well as highlighting the research area where the BFRS was determined. Also, authors should emphasize more explicit research results. In this way, readers will be given more precise information that will attract them to read the article.
Introduction: The introduction should be completed by defining the geographical areas where the research was conducted. Also, it is important that the authors point out why they have conducted research that concentrates on multi-sensor flow data obtained from smartphones, although it is known that this can be determined in a number of other ways (e.g. using vehicles specialized to detect bumps on the road, etc.). Also, it is important to point out the contribution of this study and its results.
Related Work: The authors very comprehensively refer to related works indicating the prevalence of vision-based and sensor-based methods.
Methodology: This section of the article is very extensive and describes in detail every step of the research conducted. The proposal is that at the very beginning of the section, the authors present the research area, and then what follows. Also, one gets the impression that the authors, by citing examples, put more focus on the general application of models and calculations. The advice is to adapt the research methodology to a personal example (аn example of this refers to more precise information regarding the creation of a CNN model of a neural network, and the like).
Experiments and Discussion: The authors have detailed the implementation of the research and the obtained results, however, what is not the most clear is how the authors determined the division of data sets for training and testing. Did the authors try to get perhaps better results with a different distribution (eg. 70 - 30), because the question of underfitting/overfitting arises, which the authors did not explicitly point out? Also, given that the results and discussion are combined in one chapter, no comparison of the obtained results with other studies can be seen anywhere. This would significantly improve the article and highlight the importance of the study conducted.
Conclusions and Future Work: The paper's conclusion is very concise and looks at the goal and subject of the research, emphasizing the most important results and the direction of future research.
Author Response
Title: The title of the article clearly defines the subject of the article.
Response: Thanks for your professional comments, and we have made detailed revisions and improvements for the manuscript based on your comments.
Abstract: The abstract clearly indicates the goal and subject of research based on the Bump Features of the Road Surface. However, a proposal for improving the work refers to adding information about the data sample as well as highlighting the research area where the BFRS was determined. Also, authors should emphasize more explicit research results. In this way, readers will be given more precise information that will attract them to read the article.
Response: Thank you very much for your suggestions. We have revised the abstract to include information about the data sample and to highlight the research area where the BFRS was determined. Additionally, we have emphasized more explicit research results to provide readers with precise information that will enhance their interest in reading the article.
Related descriptions have been revised the in the manuscript.
“With the sampling rate of GPS as 1 Hz, gyroscope and accelerometer as 100 Hz, a multi-sensor stream data is recorded at three different urban areas of Nanjing, China, using the smartphone mounted on a vehicle.”
“ BFRS detection experiments using multi-sensor stream data from smartphones are conducted, and 4, 14, 17 BFRS are correctly detected in three different areas, with the precision as 100%, 70.00%, and 77.27%, respectively. Then comparisons are conducted between the proposed method and three other methods, and the F-Score of the proposed method are computed as 1.0000, 0.6363, 0.7555 at three different areas, which hold highest value among all results. Finally, it shows that the proposed method performs well in different geographic areas.”
Introduction: The introduction should be completed by defining the geographical areas where the research was conducted. Also, it is important that the authors point out why they have conducted research that concentrates on multi-sensor flow data obtained from smartphones, although it is known that this can be determined in a number of other ways (e.g. using vehicles specialized to detect bumps on the road, etc.). Also, it is important to point out the contribution of this study and its results.
Response:
Thank you very much for your suggestions. We have updated the Introduction section to include the geographical areas where the research was conducted. We have also clarified why we chose to focus on multi-sensor flow data obtained from smartphones, despite the availability of other methods. Additionally, we have highlighted the contributions of this study and its results.
Related descriptions have been revised the in the manuscript.
“The widespread adoption of smartphones has transformed these devices into powerful tools for crowdsourced data collection on BFRS, which can improve the efficiency of BFRS detection operations and combine user convenience with extensive data coverage”
“To address these challenges and detect BFRS using the widely used motion sensor, this article concentrates on multi-sensor stream data obtained from smartphones positioned in arbitrary poses”
“Rather than the numerical measurement of acceleration data from multiple smartphones or varying vehicle types, the contributions of this article are (1) the modeling of vehicle movement than can deal with the BFRS in different frequency and driving condition, along with the augmentation of the training dataset; (2) the representing and detecting of BFRS in account with the noisy sampling; (3) the presentation and integration of BFRS locations based on the weighted clustering method.”
“we conduct experiments at different geographic areas in urban area of Nanjing, China. A part of the dataset is used as the training dataset, and others are used as evaluation dataset, then comparisons are conducted between the proposed method and three different methods.”
Related Work: The authors very comprehensively refer to related works indicating the prevalence of vision-based and sensor-based methods.
Response: Thanks.
Methodology: This section of the article is very extensive and describes in detail every step of the research conducted. The proposal is that at the very beginning of the section, the authors present the research area, and then what follows. Also, one gets the impression that the authors, by citing examples, put more focus on the general application of models and calculations. The advice is to adapt the research methodology to a personal example (аn example of this refers to more precise information regarding the creation of a CNN model of a neural network, and the like).
Response: Accepted.
We have revised the content of our manuscript accordingly. At the very beginning of the Methodology section, we now present the research area followed by the subsequent steps. Regarding the description of the method models, we have made explanations from a theoretical perspective on the characteristics of the proposed method in this paper, as well as the differences between this method and voice signal processing, focusing on the general application and movement modeling. The proposed method can “hear” the voice of the road surface, and detect BFRS from the input dataset. Additionally, we have adapted the research methodology to include a personal example (refers to the experiment area of Nanjing, China), providing more precise information regarding the creation and computation of the CNN neural network model.
Related descriptions have been revised the in the manuscript.
“Based on the analysis of related works, it intends to design the BFRS detection method using smartphones. With the multi-sensor stream data collected at three different urban areas in Nanjing, China, experiments are going to conducted based on the proposed method. This section introduces the methodological framework and implementation details for BFRS detection.”
“Using smartphone sensor data, particularly acceleration data for BFRS detection, is akin to "listening" to road surface changes through an accelerometer, much like how a song records information about its environment. However, unlike voice signals where a speaker’s speed remains relatively stable, vehicle speed and acceleration fluctuate in real-time according to actual road conditions. This variability can cause the same signal to manifest in multiple ways. Consequently, one of the innovations of the proposed method is on analyzing and modeling the relationship between BFRS characteristics and vehicle speed, and on enhancing features based on this relationship to accurately capture BFRS information under various speed conditions.”
“The 3-axis acceleration data is initially encoded as spectrum feature in dimensions of 30*25*3, and converted to the size of 30*25*12 by the first round of convolution operations, followed by the batch normalization. Then the process is repeated for the other four times with the dimension becomes 15*13*24, 8*7*48, 4*4*48, 4*4*48, followed by the dropout, fully connection, softmax operations with dimension finally becomes 1*1*2. Through these operations, the CNN model can “hear” the voice of the road surface, and detect BFRS from the input dataset.”
Experiments and Discussion: The authors have detailed the implementation of the research and the obtained results, however, what is not the most clear is how the authors determined the division of data sets for training and testing. Did the authors try to get perhaps better results with a different distribution (eg. 70 - 30), because the question of underfitting/overfitting arises, which the authors did not explicitly point out? Also, given that the results and discussion are combined in one chapter, no comparison of the obtained results with other studies can be seen anywhere. This would significantly improve the article and highlight the importance of the study conducted.
Response: Accepted.
Due to the geographic coordinates inherent in our detection results, which differ from traditional methods of evaluating neural network models, we opted to use data from a specific spatial area as the training set. By setting training epochs and convergence parameters, we aimed to achieve the best training outcomes for our model. For validating the effectiveness of the model and comparing it with other methods, we used data from a different area (not included in the training set) for testing and made comparisons with other approaches.
Besides, comparisons are conducted between the proposed method and three different BFRS detection methods, at the test area of Area A, and the total area of Area B and Area C. In assessing the experimental results, considering potential time deviations in BFRS detection and GPS positioning accuracy, we used a 10-meter search range as our standard for comparison. We clustered the detected BFRS trajectory points, using the cluster centers to represent BFRS locations. We then compared these to the actual BFRS positions, considering detections within 10 meters as accurate. This criterion was used as an evaluation metric to calculate precision, recall, and F1-score, and to compare the experimental outcomes of different BFRS detection methods.
We have revised the manuscript content to include an explicit declaration of the division of datasets for training and testing, detailed in Section 4.1.
Related descriptions have been revised the in the manuscript.
“Due to the geographic coordinates inherent in the detection results, which differ from traditional methods of evaluating neural network models, data from a specific spatial area was selected as the training set, and data from a different area (not included in the training set) was used for testing. In Area A, the dataset is divided into a training dataset (outside the black dotted rectangle area) and a test dataset (inside the black dotted rectangle area, i.e., Area A),”
“After that operation, the cluster result is compared with the actual BFRS positions, treating detections within 10 meters of the actual BFRS as correct, and those outside this range as wrong. These criteria serve as our evaluation metrics, and precision, recall, and F1-score are calculated. Based on the criteria, comparisons are conducted with different BFRS detection methods, in addition, BFRS in different areas are computed and results are evaluated with different methods.”
“In the result, the BFRS within the distance of 10 m is considered to be the correct BFRS and the one out of the distance is taken as wrong BFRS.”
Conclusions and Future Work: The paper's conclusion is very concise and looks at the goal and subject of the research, emphasizing the most important results and the direction of future research.
Response: Thanks.
Thank you very much for your professional comments, which have significantly enhanced the quality of our manuscript. We have conscientiously revised it accordingly.
Round 2
Reviewer 2 Report (New Reviewer)
Comments and Suggestions for Authors
The authors correctly explained many of the issues raised in review no. 1.
The only discrepancy is the information regarding signals from acceleration sensors and gyroscopic sensors.
Most likely, the system implemented in the phone itself is equipped with a signal filter. Therefore, I am not convinced that the measurement results collected from acceleration sensors are not pre-filtered.
In the case of gyroscopes, the calibration method for piezoelectric systems consists in indicating the maximum and minimum values ​​and dividing the range to indicate zero. This calibration method is commonly used in flying models, and it is performed quite often at defined time intervals. This is crucial in the spatial positioning of the flying model. For a car and straight-ahead driving, the coordinate system can be limited to a flat system, however, after about 60 seconds of measurement, the values ​​from the sensors start to flow. Hence my question about the frequency of calibrating gyroscopic sensors.
You received small values ​​of acceleration "pulses" when driving over an obstacle. This means quite high damping of the tires and suspension. In this case, it would be worth looking at the Analysis of Force While Overcoming an Obstacle.
The work is interesting and shows great potential for further research.
Author Response
The authors correctly explained many of the issues raised in review no. 1.
Response: Thanks for your professional comments which have significantly enhanced the quality of the manuscript.
The only discrepancy is the information regarding signals from acceleration sensors and gyroscopic sensors.
Response: Thanks for your insightful feedback and detailed analysis. We have made a review about the motion sensor in smartphone (here, we just take the iPhone as an example), including the acceleration sensor and gyroscopic sensor, and detailed discussions are as follows.
Most likely, the system implemented in the phone itself is equipped with a signal filter. Therefore, I am not convinced that the measurement results collected from acceleration sensors are not pre-filtered.
Response: Accepted. Regarding your concerns about the signal filtering in smartphone sensors, we acknowledge that many modern smartphones are indeed equipped with basic filtering mechanisms to improve sensor data usability. However, the degree of pre-filtering can vary significantly between devices and manufacturers. Here, we just take the iPhone as an example (since it is used as the data collecting device in the experiment). The detailed review process includes:
(1) We visited the introduction page of the MATLAB Mobile app at the AppStore and the webpage (https://www.mathworks.com/products/matlab-mobile.html), and it just said the app can acquire the sensor data (acceleration, orientation, and location, etc). Then followed by support packages of Android and iOS, we investigated the help document of accellog (https://www.mathworks.com/help/matlab/ref/mobilesensor.internal.mobiledev.accellog.html) and orientlog
(https://www.mathworks.com/help/matlab/ref/mobilesensor.internal.mobiledev.orientlog.html), and even about the AHRS filter (https://www.mathworks.com/help/nav/ref/ahrs.html, https://www.mathworks.com/help/nav/ug/estimate-orientation-using-ahrs-filter-and-imu-data-in-simulink.html). We found that there is no explicit description of the related filter technique about the data acquired by MATLAB;
(2) Then, we visited the help document of the Apple Core Motion framework (https://developer.apple.com/documentation/coremotion), which could “Process accelerometer, gyroscope, pedometer, and environment-related events”. We found that the acquired sensor data can be managed by the CMMotionManager (https://developer.apple.com/documentation/coremotion/cmmotionmanager), and get the import clue “Core Motion’s sensor fusion algorithms provide this data”. Hence, we believe the data acquired via the MATLAB Mobile app can be the “processed” data using the “sensor fusion algorithm”, based on the format of the acquired data.
(3) Following this clue, we’d like to get the specific algorithm about the “sensor fusion algorithm”, then we visited the web document of “Understanding Core Motion” (https://nonstrict.eu/wwdcindex/wwdc2012/524/). Here, we had learned much about the nature of the data collecting process by the iOS, and the acquired motion data including acceleration, orientation, were generated based on the fusion of accelerometer, gyroscope, and magnetometer. However, there is no detailed implementation about the “sensor fusion algorithm”, but we believe there can be data filtering and fusing algorithms applied before the sensor data recorded by MATLAB Mobile app.
(4) We also contacted the technical expert of Apple, however, we couldn't get the detailed information about the filter algorithm and the detailed motion fusion operation, due to some reasons as classified commercial secrets.
Finally, it can be concluded that: (1) there exists “data flow/drift” for motion sensors (i.e., the error grows over time), including the gyroscope; (2) the “sensor fusion algorithm” provided by Apple can convert the “dirty data” (with bias or error) into “clean data”; (3) there are filtering techniques or algorithms applied for the raw acceleration, then the data are recorded by MATLAB Mobile app and further sent to the proposed method.
We have revised our related section to include a more detailed explanation on the nature of data collection and inherent filtering process applied by the device.
“the data preprocessing algorithms provided by smartphone operating systems, such as the “sensor fusion algorithm” in the iPhone’s Core Motion framework, can transform this “dirty data” into “clean data.””
“Consequently, the collected dataset, including acceleration and orientation, is preprocessed and filtered. ”
“Smartphones were used as the source for sensor data collection, utilizing the inherent data processing techniques of the smartphone operating system.”
“The raw acceleration data are preprocessed using these techniques before being recorded by the MATLAB Mobile app and subsequently sent to the proposed method.”
In the case of gyroscopes, the calibration method for piezoelectric systems consists in indicating the maximum and minimum values and dividing the range to indicate zero. This calibration method is commonly used in flying models, and it is performed quite often at defined time intervals. This is crucial in the spatial positioning of the flying model. For a car and straight-ahead driving, the coordinate system can be limited to a flat system, however, after about 60 seconds of measurement, the values from the sensors start to flow. Hence my question about the frequency of calibrating gyroscopic sensors.
Response: Accepted. (1) For the calibration of gyroscopic sensors, your observation about the use of a calibration method typical for piezoelectric systems in flying models is well noted. We understand the importance of regular recalibration, especially given that sensor drift can occur, particularly over extended periods of measurement;
(2) As discussed in the above response, especially in the web document of “Understanding Core Motion” (https://nonstrict.eu/wwdcindex/wwdc2012/524/), there indeed exists “data flow/drift” for the gyroscope sensor. However, due to the “sensor fusion algorithm” provided by Apple, there is a data fusion process combined with accelerometer, gyroscope, and magnetometer. Hence, the effect of “data flow/drift” can be limited, and calibrations, such as “after about 60 seconds of measurement”, are not needed in the current experiment;
(3) We conducted the experiment using the data recorded within 10 minutes, and validated the result with the ground truth BFRS, which proved the effectiveness of the proposed method. In addition, We have further conducted an experiment that changes the spatial pose of the smartphone after a fixed time interval and record the motion data via the app, which lasted for more than 20 minutes. The result turned to be consistent with the actual pose of the smartphone.
It can be concluded that the use of smartphone with the inherent data preprocessing can be effective in dealing with the problem of “data flow/drift” for the gyroscope sensor.
We have made explicit explanations and revised the related descriptions in the manuscript.
“Although “data flow/drift” occurs in motion sensors like gyroscopes, where errors accumulate over time, the data preprocessing algorithms provided by smartphone operating systems, such as the “sensor fusion algorithm” in the iPhone’s Core Motion framework, can transform this “dirty data” into “clean data.””
“Additionally, before initiating data collection, the smartphone is restarted and left idle for a period to ensure the activation of the device’s sensor calibration features.”
“This setup effectively addresses the issue of “data flow/drift” in motion sensors (such as the gyroscope). ”
“Each dataset collected typically lasts no longer than 10 minutes. Due to the robust preprocessing capabilities of the smartphone, extra calibration for the motion sensors is not conducted during the data collection process.”
You received small values of acceleration "pulses" when driving over an obstacle. This means quite high damping of the tires and suspension. In this case, it would be worth looking at the Analysis of Force While Overcoming an Obstacle.
Response: Accepted. (1) In the previous discussion (“In the Z-axis direction, the changes in acceleration were quite noticeable, typically fluctuating above ±3m/s²”), that’s just the referenced value of 3m/s², and there are also larger acceleration “pulses” for different BFRS. However, using a threshold method to determine BFRS can lead to significant errors due to the noise. (2) Regarding the small values of acceleration "pulses" observed when driving over obstacles, we appreciate your suggestion to explore the Analysis of Force While Overcoming an Obstacle. We agree that this could provide deeper insights into the interaction between the vehicle’s tires, suspension, and road surface, which could be crucial for encoding the BFRS and interpreting the results more accurately.
Due to the limited experiment environment and measuring equipment in the current research, we plan to incorporate this analysis in our future work to enhance the robustness and applicability of our findings.
We have made explicit explanations and revised the related descriptions in the manuscript.
“Besides, for vehicles with different types of tires and suspension systems, it’s essential to explore the analysis of forces while overcoming an obstacle. This could provide deeper in-sights into the interaction between the vehicle’s tires, suspension, and road surface, which could be crucial for encoding the BFRS and interpreting the results more accurately.”
The work is interesting and shows great potential for further research.
Response: Thank you for your encouraging feedback. We are pleased to hear that our work has been found interesting and holds potential for further research. We are committed to continuing our investigations in this field and appreciate your support.
Reviewer 3 Report (New Reviewer)
Comments and Suggestions for Authors
I have no further requests for manuscript improvement.
Author Response
I have no further requests for manuscript improvement.
Response: Thanks for your professional comments which have significantly enhanced the quality of the manuscript.
This manuscript is a resubmission of an earlier submission. The following is a list of the peer review reports and author responses from that submission.
Round 1
Reviewer 1 Report
Comments and Suggestions for Authors
The research addressed the detection of bumps through several sensors and spectrum analysis. The results are satisfactory and there is no evidence of effective advantages of the use of the proposed methods regarding the existing ones.
My first concern regards the lack of knowledge on road-related concepts which is manifest on the first 4 pages of the document.
Surface distresses or defects are quite hard to identify and require the use of cutting-edge technology. The easiest one is roughness or “big irregularities” such as the bumps, as detection can make use of devices as the ones used in this work. Nevertheless, data processing and accuracy of the results is completely different from the one acceptable for the proposed study.
In this context, I recommend the authors to review completely the introduction and the related work focusing on their aim – bump detection and perhaps potholes (which are usually single discontinuities into the pavement, opposed to the bumps). As the possible contribution of the work is only methodological, the methodological needs and advantages must be exposed and discussed based on the literature review.
Provide references in the methodology and, most importantly, in the results discussion.
Are all the BFRS bumps, such as the one shown in Figure 2, or are surface defects also included in the testing sites? If yes, which other defects were included? Provide photos.
Highlight what we can learn from the application of each comparison method. Otherwise, the exercise is useless. Compare this with similar results from the literature.
Regarding conclusions, line 756, what do you want to say? That your method provides data complementary to road maintenance information?
If you are claiming this, I agree. If not, I'm afraid I have to disagree. Please review the sentence and share your ideas in other words to avoid misunderstandings.
Comments on the Quality of English LanguageThere are several small language mistakes.
Author Response
Comments and Suggestions for Authors:
Review 1:
The research addressed the detection of bumps through several sensors and spectrum analysis. The results are satisfactory and there is no evidence of effective advantages of the use of the proposed methods regarding the existing ones.
Response:
Thanks for your insightful comments, and we have revised the related contents.
Comments: My first concern regards the lack of knowledge on road-related concepts which is manifest on the first 4 pages of the document.
Response:
Accepted. The content of the paper, especially the first four pages, has been revised. Relevant concepts and expressions have been added, and the Introduction and Review sections have been updated and summarized. Comparative analyses of the methods have been conducted, and the related section (section 4.4, 4.5) has been revised.
Comments: Surface distresses or defects are quite hard to identify and require the use of cutting-edge technology. The easiest one is roughness or “big irregularities” such as the bumps, as detection can make use of devices as the ones used in this work. Nevertheless, data processing and accuracy of the results is completely different from the one acceptable for the proposed study.
Response:
Thanks for your insightful comments, and we have revised the related contents.
Comments: In this context, I recommend the authors to review completely the introduction and the related work focusing on their aim – bump detection and perhaps potholes (which are usually single discontinuities into the pavement, opposed to the bumps). As the possible contribution of the work is only methodological, the methodological needs and advantages must be exposed and discussed based on the literature review.
Response:
Accepted. The Introduction and Review sections have been revised. The characteristics and contributions of the methods used in this paper have been highlighted and emphasized.
Comments: Provide references in the methodology and, most importantly, in the results discussion.
Response:
Accepted. References to the methodology have been provided, and the relevant methods have been discussed in the results discussion (section 4.4, 4.5). Besides, the related descriptions have also been updated.
Comments: Are all the BFRS bumps, such as the one shown in Figure 2, or are surface defects also included in the testing sites? If yes, which other defects were included? Provide photos.
Response:
Thanks for your insightful comments. The BFRS in this manuscript refers to bump features of the road surface, including speed breakers, and transverse gutters, potholes, cracks, and related photos have been provided in this revision.
Comments: Highlight what we can learn from the application of each comparison method. Otherwise, the exercise is useless. Compare this with similar results from the literature.
Response:
Accepted. Detailed comparisons and explanations of different methods have been highlighted in Related works. Besides, the sources of the comparative methods have been provided, and detailed comparisons and explanations of different methods have been conducted and highlighted in Experiments.
Comments: Regarding conclusions, line 756, what do you want to say? That your method provides data complementary to road maintenance information?
If you are claiming this, I agree. If not, I'm afraid I have to disagree. Please review the sentence and share your ideas in other words to avoid misunderstandings.
Response:
Accepted. The understanding is right. The related content of the conclusions has been revised to eliminate ambiguities and further modifications have been made throughout the manuscript. We have also improved the clarity and flow of the language throughout the document (detailed revisions can be found by the Compare mode of MS Office Word).
Reviewer 2 Report
Comments and Suggestions for Authors
The authors in this paper propose a BFRS detection method based on spectrum modelling and multi-dimensional features base on the smartphone mounted on a vehicle. However, there are some weaknesses that the authors must address before publication.
- The authors claim that there are a lot of research in this direction. However, the significance and novelty of their research are not clearly articulated.
- In the paper absent information about design of mounted smartphone on a vehicle. Also, good idea to add installation and placed of smartphone on a vehicle.
- Must be added information about external envelopment and impact on the result different disturbance (vibration, magnetic fields and other). How provided calibration for multisensory of smartphone.
- All symbols in the formulas must be checked and correct, for example after formula 7 ”vk”.
My best regards,
Comments on the Quality of English LanguageMinor editing of English language required
Author Response
Comments and Suggestions for Authors:
Review 2:
The authors in this paper propose a BFRS detection method based on spectrum modelling and multi-dimensional features base on the smartphone mounted on a vehicle. However, there are some weaknesses that the authors must address before publication.
Response:
Thanks for your insightful comments, and we have revised the related contents.
Comments: - The authors claim that there are a lot of research in this direction. However, the significance and novelty of their research are not clearly articulated.
Response: Accepted. The significance and novelty of these methods have been highlighted and emphasized, besides, the related content in Introduction and Review sections have been revised.
Comments: - In the paper absent information about design of mounted smartphone on a vehicle. Also, good idea to add installation and placed of smartphone on a vehicle.
Response: Thanks for your insightful comments, and the relevant sections (section 3.1) of the paper have been revised. Additionally, the smartphone can be placed or installed in any position, since the orientation sensor allows for real-time calculations. This has been explained in the relevant sections of the paper.
Comments: - Must be added information about external envelopment and impact on the result different disturbance (vibration, magnetic fields and other). How provided calibration for multisensory of smartphone.
Response:
Thanks for your insightful comments. Since it utilizes a smartphone, the sensors within the phone have already pre-calibrated, besides, the impact of the sensors operating at short intervals, such as 100Hz, on the results is limited compared with the vehicle speed ranging from 0 km/h to 60 km/h. Therefore, linear interpolation techniques have been employed to make a time alignment for the collected data of multiple sensors. Overall, we have made an explanation about the external envelopment and impact on the result different disturbance. In IoT sensor application scenarios, calibrating sensors is essential. it provides constructive guidance for future practical implementations. Therefore, we have discussed this issue in the Conclusion section.
In addition, at the system level, calibrations of multiple sensors of smartphones can be made through the system settings (for example, in iOS, settings can be conducted via “System”->”Privacy”->”Location Services”->”System Services”->“Compass Calibration” and “Motion Calibration & Distance”)
Comments: - All symbols in the formulas must be checked and correct, for example after formula 7 ”vk”.
Response:
Accepted. The related formulas have been revised. Additionally, we have made a full check in the paper and detailed revisions have been made to all the formulas, letters, and other expressions throughout the text. We have also improved the clarity and flow of the language throughout the document (detailed revisions can be found by the Compare mode of MS Office Word).
Thank you again.
Round 2
Reviewer 2 Report
Comments and Suggestions for Authors
Paper was corrected in line with review. It can be accepted in the present form.